# Specific exercise patterns generate an epigenetic molecular memory window that drives long-term memory formation and identifies ACVR1C as a bidirectional regulator of memory in mice

Ashley A. Keiser [1,2,3], Tri N. Dong [1,2,3], Enikö A. Kramár[1,2,3], Christopher W. Butler[3,4], Siwei Chen[5], Dina P. Matheos[1,2,3], Jacob S. Rounds[1,2,3], Alyssa Rodriguez[1,2,3], Joy H. Beardwood [1,2,3], Agatha S. Augustynski[1,2,3], Ameer Al-Shammari[1,2,3], Yasaman Alaghband[1,2,3], Vanessa Alizo Vera [1,2,3], Nicole C. Berchtold[3,4], Sharmin Shanur[1,2,3], Pierre Baldi [5], Carl W. Cotman[3,4] & Marcelo A. Wood[1,2,3] ✉

Exercise has beneficial effects on cognition throughout the lifespan. Here, we demonstrate that specific exercise patterns transform insufficient, subthreshold training into long-term memory in mice. Our findings reveal a potential molecular memory window such that subthreshold training within this window enables long-term memory formation. We performed RNA-seq on dorsal hippocampus and identify genes whose expression correlate with conditions in which exercise enables long-term memory formation. Among these genes we found *Acvr1c*, a member of the TGF ß family. We find that exercise, in any amount, alleviates epigenetic repression at the *Acvr1c* promoter during consolidation. Additionally, we find that ACVR1C can bidirectionally regulate synaptic plasticity and long-term memory in mice. Furthermore, *Acvr1c* expression is impaired in the aging human and mouse brain, as well as in the 5xFAD mouse model, and over-expression of *Acvr1c* enables learning and facilitates plasticity in mice. These data suggest that promoting ACVR1C may protect against cognitive impairment.

The ability to learn, consolidate and retrieve information is critical for everyday life. Cognitive function begins to decline with normal aging[1–3] and is severely exacerbated by Alzheimer's Disease (AD)[4]. Basic research and clinical trials universally demonstrate the benefits of exercise for cognitive function. Exercise benefits include enhancements in long-term memory formation in humans and animals[5–9], enabling learning under inadequate, subthreshold training conditions[10–13] in mice and enhancing synaptic plasticity[13,14] in both humans and mouse models, and reducing the risk of cognitive decline associated with normal aging and AD[9,15–18]. However, age and physical

[1]Department of Neurobiology and Behavior, University of California Irvine, Irvine, CA 92697, USA. [2]Center for the Neurobiology of Learning and Memory (CNLM), University of California, Irvine, Irvine, CA 92697, USA. [3]Institute for Memory Impairments and Neurological Disorders (UCI MIND), University of California, Irvine, Irvine, CA 92697, USA. [4]Department of Neurology, University of California Irvine, Irvine, CA 92697, USA. [5]Institute for Genomics and Bioinformatics, School of Information and Computer Science, University of California, Irvine, Irvine, CA 92697, USA. ✉e-mail: mwood@uci.edu

disability-related factors that may affect individuals throughout the lifespan progressively limit and reduce engagement in exercise and associated cognitive benefits[19]. Physical activity induces the neuroprotective growth factor, Brain Derived Neurotrophic Factor (BDNF) in the cortex and hippocampus[20]. The ability for exercise to enable and drive learning in conditions that are normally subthreshold for encoding is dependent on epigenetic-mediated up-regulation of BDNF in the hippocampus[10]. Notably, BDNF benefits are maintained within a particular window of time as BDNF is re-induced with a brief, 2-day exercise session following a break from physical activity, demonstrating that a molecular memory window exists for BDNF induction[21]. In recent studies, we determined that a period of initial exercise also creates and maintains a molecular memory window for exercise benefits on cognitive function in male[11] and female[22] mice where a brief, 2-day exercise session following a break can also re-engage cognitive benefits, enhance long-term potentiation, and allow for learning under insufficient, subthreshold training conditions. Importantly, subthreshold training (that does not lead to long-term memory formation) occurring during this molecular memory window is transformed into long-term memory, suggesting molecular and epigenetic mechanisms engaged and maintained in the molecular memory window are uniquely important for the formation of long-term memory.

Here, we build on these exercise parameters to begin to define a mechanism responsible for maintaining cognitive benefits underlying this molecular memory window by initial exercise and driving long-term memory formation. We utilized an unbiased RNA-sequencing approach to uncover genes in the dorsal hippocampus that are differentially expressed under conditions where exercise benefits are maintained throughout sedentary delay periods and enable the formation of long-term memory and synaptic plasticity. As training is insufficient for long-term memory encoding in sedentary mice (with the approach used in this study), identifying differential expression of genes during consolidation when exercise facilitates learning and maintains cognitive benefits will provide insight for molecular gatekeepers and drivers of long-term memory formation.

We identify a gene coding for a type 1 activin A membrane receptor kinase of the TGF-β family of signaling molecules, *Acvr1c* as one of few genes showing up-regulation in conditions where exercise enabled the formation of long-term memory under inadequate, subthreshold learning conditions, indicating a gate-type role for *Acvr1c* in driving memory consolidation. ACVR1C belongs to the type I TGF-β receptor group, which interact with type II TGF-β receptors at the membrane to carry out TGF-β family-dependent signaling via interaction with SMAD2/3/4 complexes[23]. SMAD2/3/4 complexes can facilitate transcription by engaging histone acetyltransferases such as CREB-binding protein, (CBP)[24]. A role for type I receptors in synaptic tagging has been reported[25] and other reports highlight the importance of ligands that target TGF-β receptors in hippocampus-dependent synaptic plasticity and spatial-related memory requiring the hippocampus[26,27]. However, whether exercise engages TGF-β signaling and is maintained to facilitate cognitive function is unknown. Further, the role of individual TGF-β receptor subtypes on hippocampus-dependent memory and plasticity is unclear as manipulations specifically targeting ACVR1C or other TGF-β receptor subtypes have not been published to date. Therefore, the selection of ACVR1C for further assessment of its role in hippocampus-dependent memory and synaptic plasticity is motivated by its transcriptional regulatory role through SMAD-mediated signaling. Additionally, the inherent druggability and literature supporting a role for type I receptors in cognitive function and synaptic plasticity poise ACVR1C as a promising target. Lastly, because the SMAD proteins interact with chromatin modifying enzymes, like CBP, it may be that ACVR1C may participate in the signaling and mechanisms that ultimately modify the epigenome in response to exercise making it a desirable target from our perspective.

Misregulation of the TGF-β1 pathway is a characteristic phenotype in the hippocampus of patients with Alzheimer's Disease[28–35] and occurs in the aging hippocampus in both human and animal models[34,35]. Thus, in this study, we also examine the regulation of *Acvr1c* in the aging brain and in a mouse model of AD and determine whether manipulating *Acvr1c* expression enables learning and enhances synaptic plasticity.

In summary, we identify exercise parameters capable of maintaining cognitive benefits of exercise and enabling memory formation to occur under subthreshold training conditions. Applying these parameters, unbiased bulk RNA-Seq, ChIP-qPCR, and targeted viral manipulations, we find that exercise enhances cognitive function by engaging an epigenetic molecular memory window and driving *Acvr1c* to facilitate memory formation and synaptic plasticity in the adult brain. Further, we find that overexpression of wildtype *Acvr1c* enables learning in aging mice and in the 5xFAD mouse model of AD. Bidirectional regulation of memory formation and synaptic plasticity by ACVR1C and rescue of age and AD-associated impairments has important implications for the design of therapeutics for age- and AD-related cognitive dysfunction.

## Results
### Exercise establishes a molecular memory window for engaging and maintaining brain benefits

To examine whether specific patterns of exercise benefit the hippocampus and allow for changes in gene expression and enhancement of long-term memory formation and synaptic plasticity, mice were put through an exercise protocol that we previously found to enhance cognitive function through initial exercise, be maintained during sedentary delays and can become reactivated by re-engagement in brief, subthreshold reactivating exercise[10–12,22]. The protocol consists of 14 days of voluntary wheel-running followed by a 7- or 14-day sedentary delay and a brief, subthreshold 2-day reactivating exercise session (Fig. 1A). 14 days of exercise robustly facilitates long-term memory by enabling learning under a subthreshold version of the hippocampus-dependent Object Location Memory (OLM) task as we have demonstrated in several previous studies[10,11,22]. 14 days of exercise followed by a 7-day sedentary interval results in decreased memory performance, however a brief 2-day subthreshold reactivating exercise session results in enhanced memory performance (Fig. 1B; data taken from our previous study Butler et al.[11]; see Butler et al.[11] for performance data and other details). This pattern of exercise and memory benefit suggest that a molecular memory window (Fig. 1C) may exist where initial exercise benefits are maintained by mechanisms induced with initial exercise that persist for at least 7 days because the 2-day subthreshold reactivating exercise (which does not enhance memory formation by itself) leads to increased memory performance. A brief 2-day subthreshold reactivating exercise session does not significantly result in enhanced memory performance following a 14 day sedentary delay, though a visual trend towards enhancement is noted. Therefore, results from Butler et al.[11], suggest that benefits of reactivating exercise begin to fade with increasing sedentary days.

To determine how these exercise patterns affect cellular mechanisms we next examine hippocampal long-term potentiation (LTP) in dorsal hippocampal slices using theta burst stimulation (TBS), a form of synaptic plasticity thought to underlie memory formation. Similar to our behavioral results (Fig. 1B), we observe that 14 days of initial exercise leads to a significant increase in LTP relative to sedentary control (Fig. 1D). Interestingly, a 7 day sedentary delay did not affect the level of LTP (Fig. 1E), in contrast to a 14 day delay (Fig. 1F). The 2-day subthreshold reactivating exercise leads to enhanced LTP after a 14 day delay (Fig. 1F). Thus, although the LTP does not parallel the behavior exactly (although not surprising as the behavior is incidental learning and the LTP is induced by a robust stimulation

protocol), the main observation holds that specific exercise patterns induce mechanisms that persist through specific sedentary delays and facilitate memory and synaptic plasticity upon exercise re-engagement.

## Acvr1c is induced during consolidation only under exercise conditions that facilitate memory formation

To begin to understand the mechanisms defining the molecular memory window for exercise, we employ an unbiased approach using

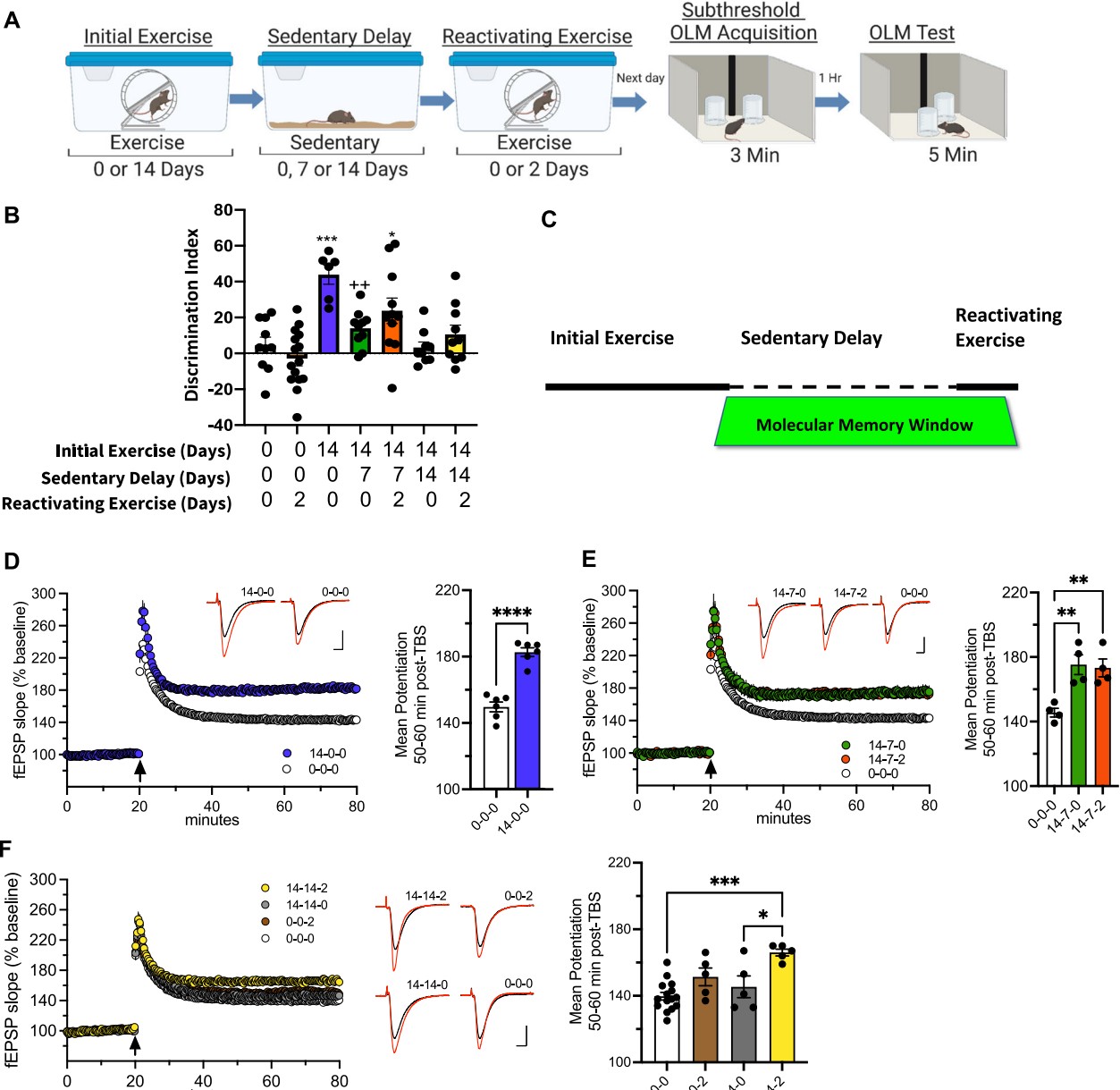

**Fig. 1 | Exercise enhances hippocampus-dependent memory and synaptic plasticity and engages a molecular memory window leading to enhanced cognitive improvement and plasticity from subsequent 2-day reactivating exercise. A** Simplified schematic; see[11]. Created with Biorender.com. **B** Discrimination Index (DI) scores during object location memory (OLM) test. 14 days of exercise transforms a normally subthreshold learning event into long-term memory. After a 7-day sedentary delay, only 2 days of reactivating exercise enables long-term memory formation. One-way ANOVA, Group: ($F_{(6,64)} = 8.13$, $P < 0.0001$). Tukey's post hoc test: *$P < 0.05$, ***$P < 0.001$ compared to sedentary (0-0-0), ++$P < 0.01$ compared to 14-0-0, (0-0-0: n = 10, 0-0-2: n = 15, 14-0-0: n = 6, 14-7-0: n = 10, 14-7-2: n = 11, 14-14-0: n = 9, 14-14-2: n = 10). Data taken from our previous study Butler et al., 2019[11]. Data are presented as mean ± SEM. **C** Exercise parameters and framework. **D** Left panel, time course of the mean ± SEM field excitatory postsynaptic potential (fEPSP) slope as a percentage of baseline recorded in slices. Right panel, independent two-sample *t* test (two-tailed): (t(10) = 8.261, $P < 0.0001$),

(14-0-0: n = 6 mice, n = 12 slices, control 0-0-0: n = 6 mice, n = 12 slices). Inset; representative traces collected during baseline (black line) and 60 min post theta burst stimulation (TBS, arrow) (red line). Scale = 1 mV/5 ms. **E** Left panel, LTP time course. Right panel, quantification of mean ± SEM potentiation 50–60 min post-TBS, One-way ANOVA, Group: ($F_{(2,9)} = 10.96$, $P = 0.0039$). Tukey's post hoc test, 50–60 min post TBS: **$P < 0.01$ for both 14-7-0 and 14-7-2 compared to 0-0-0. (14-7-0: n = 4 mice, n = 8 slices, 14-7-2: n = 4 mice, n = 8 slices, 0-0-0: n = 4 mice, n = 8 slices). Inset; representative traces collected during baseline and 60 min post-TBS. Scale = 1 mV/5 ms. **F** Left panel, LTP time course. Middle panel, representative traces. Scale = 1 mV/5 ms. Right panel, One-way ANOVA, Group: ($F_{(3,26)} = 8.942$, $P = 0.0003$). Tukey's post hoc test, 50–60 min post TBS: **$P < 0.001$ for 14-14-2 compared to 0-0-0, *$P < 0.05$ for 14-14-2 compared to 14-14-0, $P = 0.137$ for 0-0-2 compared to 0-0-0. (14-14-0: n = 5 mice, n = 10 slices, 14-14-2: n = 5 mice, n = 10 slices, 0-0-2: n = 5 mice, n = 10 slices, 0-0-0: n = 15 mice, n = 30 slices). Source data are provided as a Source Data file.

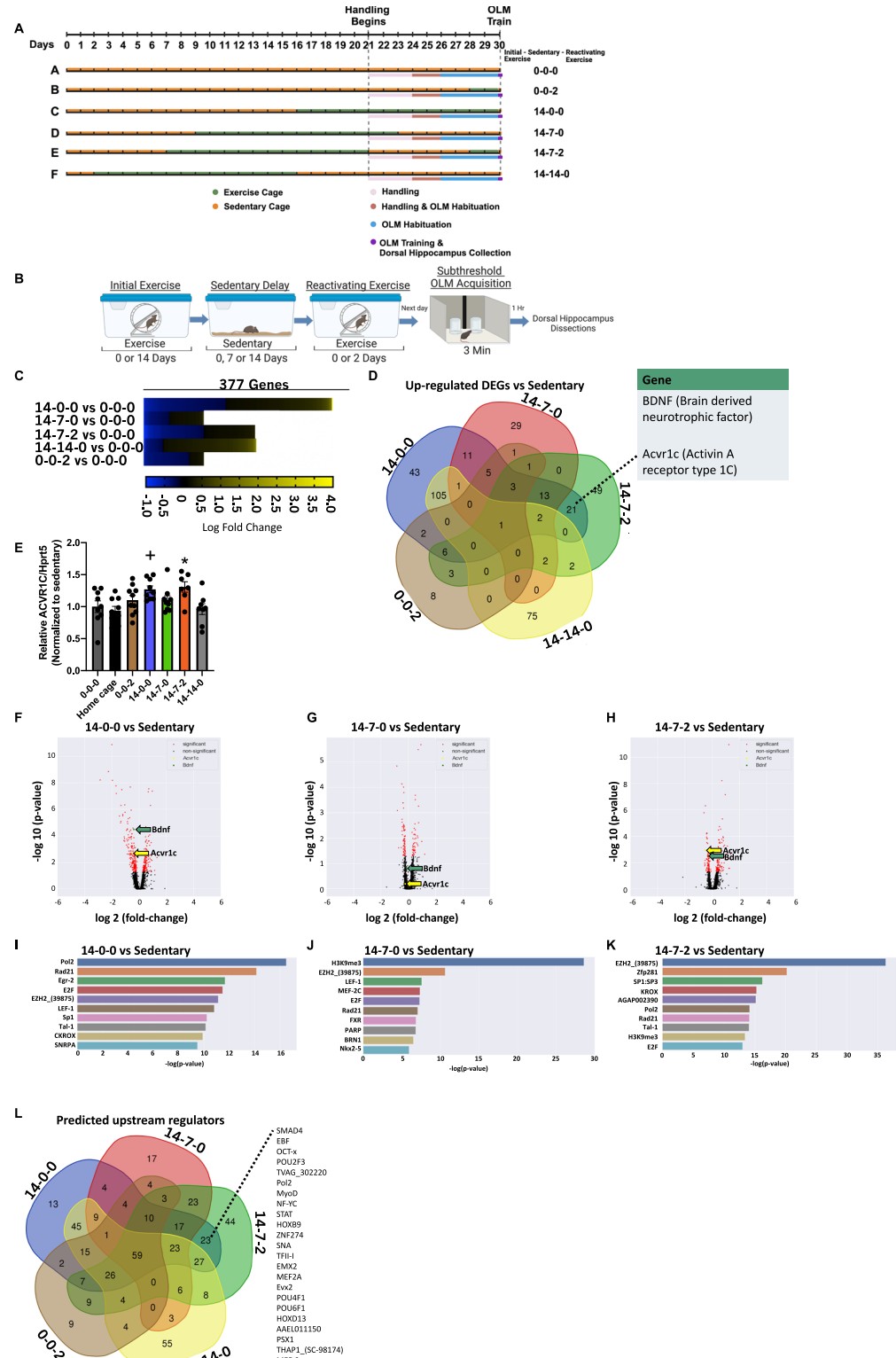

bulk tissue RNA-sequencing with dorsal hippocampus collected following different exercise parameters (Fig. 2A) 1 h post acquisition in the OLM task (as this corresponds to a main memory consolidation period) (Fig. 2B). We focus on dorsal CA1 as we have found this region to be specifically involved in object location memory[36–38]. Engagement in different exercise parameters results in a diverse transcriptional profile as defined by differences in the number, pattern, and type of genes up and down-regulated (Fig. 2C). To identify genes that play a critical role in cognitive enhancement, we next focus our attention on

exercise groups that enable robust long-term memory formation under inadequate subthreshold acquisition sessions in the OLM task (14-0-0 and 14-7-2). By intersecting up-regulated genes when exercise facilitates learning (14-0-0 and 14-7-2) relative to sedentary control (Fig. 2D), we identify potential regulators of learning and targets for memory enhancement. A small subset of 21 genes was identified (Data Table S1). This subset includes *Bdnf*, a gene we have previously demonstrated to follow a similar pattern of up-regulation in hippocampus with exercise[12,21] and studies from multiple labs have

**Fig. 2 | *Acvr1c* expression in hippocampus is induced during object location memory (OLM) consolidation only under exercise conditions that drive memory encoding. A** Detailed schematic displaying the timing of procedures for individual groups. **B** Simplified Schematic. Created with Biorender.com. **C** Heat map of genes differentially expressed in each condition compared to sedentary control (0-0-0). Positive log fold change indicates up-regulated genes vs sedentary whereas negative log fold change refers to down-regulated genes. **D** Number of up-regulated differentially expressed genes (DEGs) in dorsal hippocampus compared to sedentary. **E** RT-qPCR data demonstrating that *Acvr1c* is up-regulated in dorsal hippocampus only in conditions where exercise facilitates learning (14-0-0, n = 9 and 14-7-2, n = 7). One-way ANOVA, Group ($F_{(6,54)}$ = 3.62, P = 0.004). Tukey's post hoc test: *P < 0.05, +P = 0.05 compared to sedentary (0-0-0, n = 9). Normalization

to sedentary (HC and 14-7-0, n = 9, 0-0-2 n = 10 and 14-14-0 n = 8). Data are presented as mean ± SEM. **F–H** Overlapping volcano plots illustrating the significance (Y-axis) and magnitude (X-axis) of experience-induced changes in each group. Volcano plots show fold change in gene expression between two conditions, using regularized t test and *p* value corrected for multiple testing. *Acvr1c* and *Bdnf* were up-regulated during memory consolidation in both conditions where exercise facilitates learning (14-0-0) (**F**) and (14-7-2) (**H**). **I** Top predicted upstream regulators of 14d initial exercise (14-0-0) vs. sedentary DEGs, 7d sedentary delay (14-7-0) vs. sedentary (**J**), and 2d reactivating exercise (14-7-2) vs. sedentary (**K**). **L** Venn diagram highlighting common upstream regulators in exercise conditions that facilitate learning (14-0-0 and 14-7-2). Source data are provided as a Source Data file.

demonstrated to be required for exercise-induced cognitive enhancement[10,15,39,40], serving as a positive control in this study. From the remaining 20 genes, we focused our attention on examining the role of *Acvr1c* in long-term memory, a TGF-β1 receptor family subtype (see discussion for additional rationale). RT-qPCR results using RNA from the same samples are in line with RNA-Seq data with greater *Acvr1c* abundance in both exercise conditions that facilitate learning relative to sedentary control (Fig. 2E). In a different cohort of mice, we examine *Acvr1c* expression after exercise and subthreshold OLM training (as in Fig. 2E) or after exercise alone to determine whether exercise alone is sufficient to induce *Acvr1c* expression. Indeed, we found greater *Acvr1c* abundance following 14 days of exercise whether or not dorsal hippocampus tissue was taken during memory consolidation (Fig S1), suggesting that exercise alters gene expression (of at least *Acvr1c*) that may then subsequently affect subthreshold learning in OLM. Volcano plots display patterns of gene expression for exercise groups and highlight *Acvr1c* and *Bdnf* (Fig. 2F–H). *Acvr1c* and *Bdnf* are also identified as differentially expressed using linear regression models[41] (Fig S2) by intersecting up-regulated genes in both conditions when exercise facilitates learning (14-0-0 and 14-7-2) (Fig. S2C). As a caveat, it should be noted that the initial 14 day running distance varied across the groups used for RNA-seq (Fig. S3A). Importantly, however, up-regulation of *Acvr1c* and *Bdnf* when exercise facilitates learning is not due to greater initial running distance during the 14 day period when compared with other exercise groups as mice with a 7 day sedentary delay (where *Acvr1c* is not up-regulated: 14-7-0) run more than mice with no delay (time point when *Acvr1c* is up-regulated:14-0-0) (Fig. S3A). To test whether the initial 14 day running distance predicted relative *Acvr1c/Hprt* levels, a simple linear regression analysis was performed. The overall regression analysis did not yield statistically significant results, suggesting that the model, which incorporated running distance as the independent variable, did not achieve statistical significance. The relatively low R-squared value underscores the modest association between running distance and gene expression levels, further supporting the notion that the relationship is not strongly established based on the current model (Fig. S3B). We next identify potential regulators upstream of gene targets within the window for cognitive enhancement (Fig. 2I–K). To focus our analysis on potential upstream regulators of gene targets up-regulated when exercise facilitates learning (14-0-0 and 14-7-2; highlighted in Fig. 2D), we intersect and display predicted upstream regulators for only these groups (Fig. 2L). Following this, we use three complementary analyses to examine the mechanisms responsible for maintaining the molecular memory window for exercise utilizing up-regulated differentially expressed genes compared to sedentary controls. First, we conduct gene coexpression network analysis to identify potential hub genes for each exercise condition (Fig. S4). Second, we identify predicted pathways altered by exercise (Fig. S5). Third, we perform gene ontology enrichment analysis of cellular components altered by exercise (Fig. S6). Together, our results identify *Acvr1c* as one of few genes up-regulated only under conditions where exercise

enabled the formation of long-term memory and synaptic plasticity, suggesting a role for *Acvr1c* in facilitation of memory consolidation.

**Exercise modulates epigenetic regulation of *Acvr1c* and *Bdnf* during consolidation and reveals a specific permissive signature**
To assess how exercise modulates epigenetic regulation of *Acvr1c* during consolidation, histone modifications associated with either transcriptional activation (H3K9Ac[42,43] and H3K27Ac[44]) or repression (H3K9Me3[42] and H3K27Me3[42]) were examined at *Acvr1c* and *Bdnf IV* promoters using chromatin immunoprecipitation (ChIP-qPCR) (Fig. 3A). Occupancy at *Bdnf IV* promoter was examined given the role of this transcript in exercise-facilitated learning and abundant expression in the hippocampus[10]. We chose to present the ratio of acetylation to methylation in Fig. 3 with the individual graphs presented in supplemental data (Fig. S7). We found that exercise significantly impacts H3K9Me3/H3K9Ac at *Acvr1c* following 2 weeks of exercise (14-0-0) but does not impact promoter occupancy following sedentary delay periods (Fig. 3B, Fig. S7A, B). Exercise does not significantly impact H3K9Me3/H3K9Ac at *Bdnf IV* promoters following sedentary delay periods (Fig. 3D, Fig. S7E, F). For example, the ratio change is not maintained in the 14-7-2 group for *Acvr1c*, and no ratio differences are present across groups for *Bdnf IV* (Fig. 3B, D). In contrast, engagement in extensive (14-day) exercise reduces repressive H3K27Me3 at both *Acvr1c* (Fig. 3C, Fig. S7C, D) and *Bdnf IV* (Fig. 3E, Fig. S7G, H) promoters. Notably, this reduction and change in ratio of H3K27Me3/Ac is present during sedentary delay periods assessed and following reactivating exercise. These findings identify a specific epigenetic modification that is regulated by exercise, but that does not necessarily correlate with the LTP and long term memory results suggesting that there may be other chromatin modifications or chromatin remodeling mechanisms involved. Results provide initial evidence for the ability of exercise to effectively remove repressive marks at two distinct promoters of genes critical for memory in a specific manner, ultimately contributing to the establishment of a permissive epigenetic signature during memory consolidation.

**ACVR1C is required for hippocampus-dependent long-term memory and synaptic plasticity**
Type 1 TGFβ receptors (including ACVR1C) function as protein kinases that phosphorylate SMAD 2/3 proteins, enabling complex formation with SMAD 4 and subsequent nuclear insertion, transcriptional regulation and facilitation[23,45]. ACVR1C is abundant throughout the hippocampus, including the CA1 region targeted here (Fig. S8). Whether ACVR1C plays a role in hippocampus-dependent long term memory formation and synaptic plasticity is unknown. Therefore, to determine the requirement of *Acvr1c*, a gene coding for a type 1 receptor for the TGFβ family of signaling molecules in hippocampus-dependent long-term memory and synaptic plasticity, we designed a kinase-dead point mutant AAV construct (Fig. 4A) to disrupt the function of *Acvr1c*. Two weeks following infusions of the kinase-dead or AAV-EV control construct targeting the dorsal hippocampus, mice were handled for four

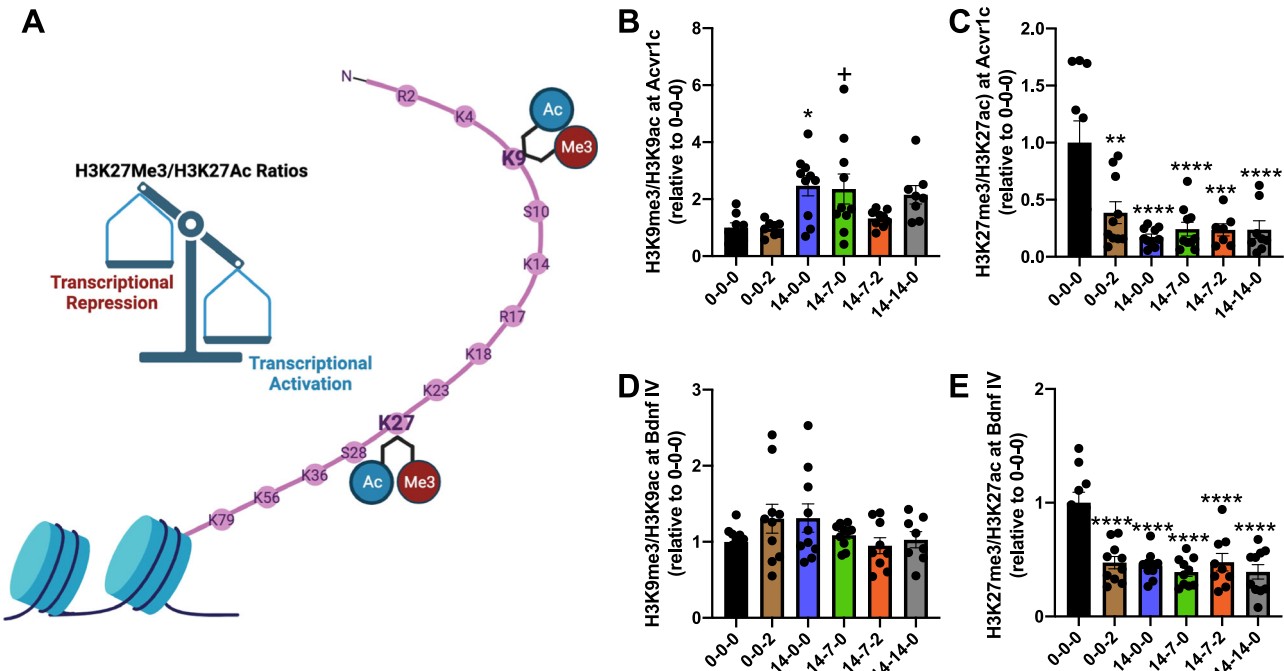

**Fig. 3 | Exercise modulates epigenetic regulation of *Acvr1c* and *Bdnf* during consolidation and reveals a specific permissive signature. A** Schematic. Created with Biorender.com. **B, C** Epigenetic regulation of *Acvr1c* during consolidation is modified by exercise. **B** Exercise does not impact H3K9Me3/H3K9Ac occupancy at the *Acvr1c* promoter following sedentary delay periods: One-way ANOVA, Group ($F_{(5,47)}$ = 4.41, P = 0.002), (0-0-0: n = 8, 0-0-2: n = 8, 14-0-0: n = 10, 14-7-0: n = 10, 14-7-2: n = 9, 14-14-0: n = 8). **C** Engagement in either minimal (2-day), (**P < 0.01) or extensive (14-day), (****P < 0.0001) exercise reduces repressive H3K27Me3 at the *Acvr1c* promoter. Notably, this reduction occurs throughout the sedentary delay periods assessed (7-day), (****P < 0.0001) and (14-day), (****P < 0.0001) and following reactivating exercise (***P < 0.001): (One-way ANOVA), Group ($F_{(5,49)}$ = 9.377, P < 0.0001), (0-0-0: n = 10, 0-0-2: n = 10, 14-0-0: n = 10, 14-7-0: n = 10, 14-7-2: n = 7, 14-14-0: n = 8). **D, E** Epigenetic regulation of *Bdnf IV* during consolidation is modified by exercise. **D** Exercise does not impact H3K9Me3/H3K9Ac

occupancy at the *Bdnf IV* promoter: One-way ANOVA, Group ($F_{(5,51)}$ = 1.44, P = 0.22), (0-0-0: n = 10, 0-0-2: n = 10, 14-0-0: n = 10, 14-7-0: n = 10, 14-7-2: n = 9, 14-14-0: n = 8). **E** Remarkably, engagement in either minimal (2-day), (****P < 0.0001) or extensive (14-day) (****P < 0.0001) exercise also reduces repressive H3K27me3 at the *Bdnf IV* promoter. Notably, this reduction also continues at the sedentary delay periods assessed (7-day), (****P < 0.0001) and (14-day), (****P < 0.0001) sedentary periods and following reactivating exercise (****P < 0.0001): (One-way ANOVA), Group ($F_{(5,53)}$ = 13.90, P < 0.0001), (0-0-0: n = 10, 0-0-2: n = 10, 14-0-0: n = 10, 14-7-0: n = 10, 14-7-2: n = 9, 14-14-0: n = 10). Results provide evidence for the ability of exercise to effectively remove repressive marks at two distinct promoters of genes critical for memory in a specific manner during consolidation. Tukey's post hoc test: *P < 0.05, **P < 0.01, ***P < 0.001, ****P < 0.0001, + P = 0.05. Data are presented as mean ± SEM. Source data are provided as a Source Data file.

days and received five-minute habituation sessions to the context over the course of six days (Fig. S9A). Importantly, mice from both treatment groups successfully habituate to the context and group differences are not observed (Fig. S9A). The day following habituation, mice were given a standard 10-minute object location memory session (which we have shown results in long-term object location memory 24 h later[36,46–53]) and memory was tested the next day by examining exploration of the object in the novel location (Fig. 4A). Disrupting the function of *Acvr1c* results in impaired performance in the object location memory test compared with the EV control (Fig. 4D). Importantly, mice do not show an object preference on training (Fig. 4B, C), exploration does not differ between groups on test (Fig. 4E) and there is no difference in anxiety between groups (Fig. S10A). To further probe the requirement of *Acvr1c*, hippocampus-dependent memory was assessed with context fear conditioning; mice were trained in context fear conditioning and memory was tested the following day. Similar to OLM, freezing is impaired relative to AAV-EV control mice that received the *Acvr1c* kinase-dead construct (Fig. 4F), suggesting that ACVR1C is required for hippocampus-dependent long-term memory. Importantly, *Acvr1c* manipulation has no effect on reactivity to the shock on training day given similar movement during shock onset (Fig. S11A). To determine whether ACVR1C is required for synaptic plasticity in dorsal hippocampal slices, LTP was examined in slices from a cohort of randomly selected mice following behavior (Fig. 4G). Figure 4H summarizes the mean potentiation of LTP during the last 10 min of recording. Disrupting ACVR1C function results in

impaired LTP relative to AAV-EV controls (Fig. 4H). Together, these results demonstrate that ACVR1C function is necessary for hippocampus-dependent long-term memory and synaptic plasticity.

## *Acvr1c* overexpression transforms inadequate, subthreshold learning into long-term memory and facilitates LTP
To determine whether ACVR1C can enhance long-term memory formation and synaptic plasticity, wildtype (WT) *Acvr1c* was overexpressed in dorsal hippocampus (Fig. 5A). 3 mo. male mice received viral infusions of either AAV-ACVR1C or AAV-EV. Two weeks later, mice were handled for four days and received five-minute habituation sessions to the context over the course of six days (Fig. S9B). Importantly, mice from both treatment groups successfully habituate to the context and group differences are not observed (Fig. S9B). The day following habituation, mice were given a 3-min subthreshold OLM acquisition session (as in[10,11,22,36,38,50,51,54]) and memory was tested the following day. *Acvr1c* overexpression (WT) in area CA1 enhances memory for object location, with AAV-ACVR1C mice showing a stronger preference for the new object location compared with AAV-EV control (Fig. 5D). Importantly, mice do not show an object preference on training (Fig. 5B, C), exploration does not differ between groups on test (Fig. 5E) and there was no difference in anxiety between groups (Fig. S10B). To probe additional aspects of hippocampus-dependent memory, mice were trained in a subthreshold context fear conditioning task[55] and tested the following day. Freezing is not enhanced in AAV-ACVR1C mice relative to AAV-EV control (Fig. 5F). Importantly, *Acvr1c* manipulation

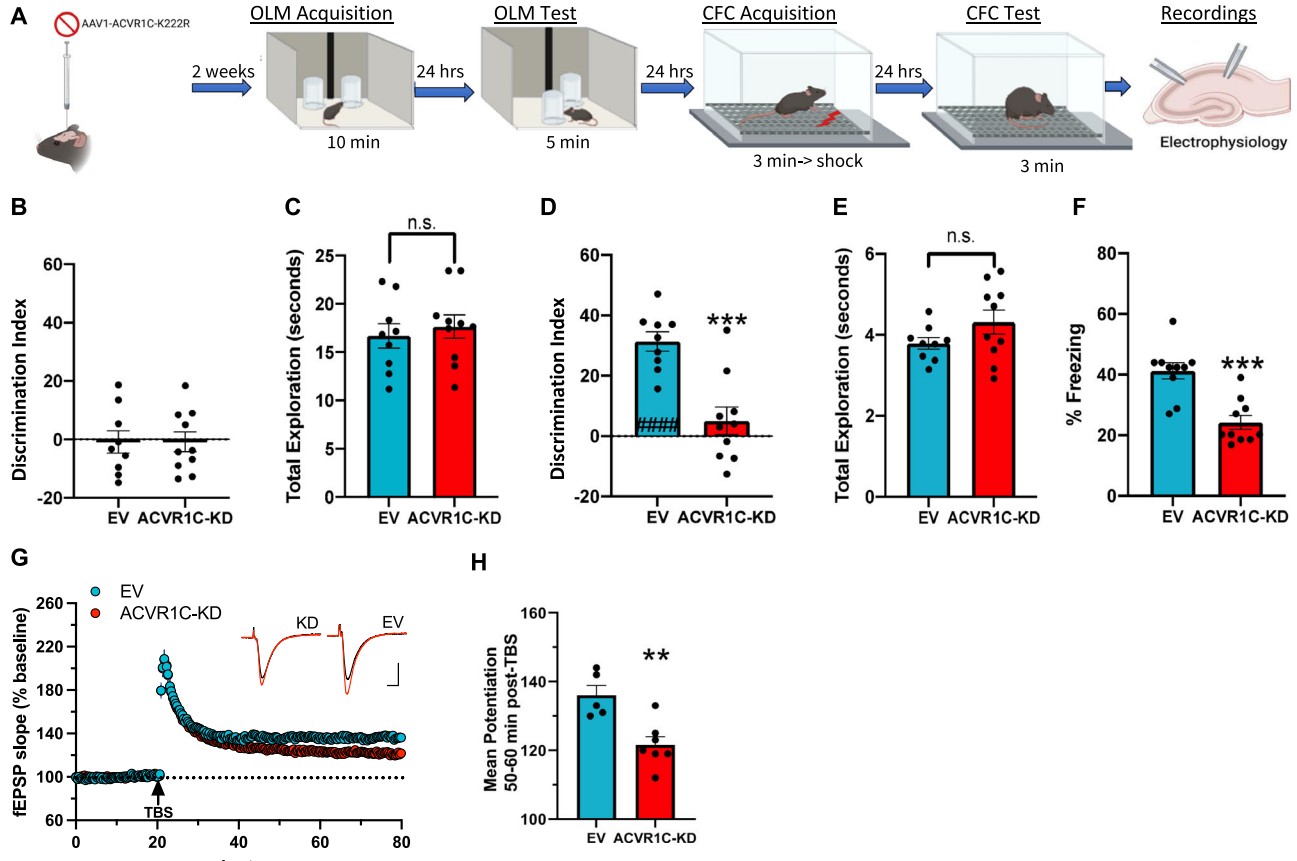

**Fig. 4 | ACVR1C is required for hippocampus-dependent long-term memory and synaptic plasticity. A** Experimental design. Created with Biorender.com. **B** Discrimination Index (DI) scores during object location memory (OLM) training reveal no differences between groups independent two-sample *t*-test (two-tailed): (t(17) = 0.006, P = 0.995), (empty vector (EV): n = 9, ACVR1C-kinase dead (ACVR1C-KD): n = 10). **C** Total amount of time in seconds exploring objects during training. Mice from both groups display similar levels of total object exploration independent two-sample *t*-test (two-tailed): (t(17) = 0.545, P = 0.592). **D** Disrupting ACVR1C function (ACVR1C-KD) leads to impaired OLM independent two-sample *t* test (two-tailed): (t(17) = 4.65, P = 0.0002), (EV: n = 9, ACVR1C-KD: n = 10). **E** Total amount of time in seconds exploring objects during test. Total object exploration does not differ between groups during test independent two-sample *t* test (two-tailed): (t(17) = 1.53, P = 0.14), (EV: n = 9, ACVR1C-KD: n = 10). **F** Percent freezing during

3-minute test session. Disrupting ACVR1C function (ACVR1C-KD) led to reduced freezing compared with EV control independent two-sample *t*-test (two-tailed): (t(18) = 4.790, P = 0.0001), (EV: n = 10, ACVR1C-KD: n = 10). **G** LTP as mean ± SEM excitatory postsynaptic potential (fEPSP) slope as percentage of baseline overtime (ACVR1C-KD; n = 7 mice, n = 14 slices, EV; n = 5 mice, n = 10 slices). Theta burst stimulation (TBS) applied at arrow. Inset; representative traces collected during baseline (black line) and 60 min post TBS (red line) from ACVR1C-KD and EV group. Scale = 1 mV/5 ms. **H** Mean ± SEM potentiation 50–60 min post TBS showing impaired LTP in slices from KD-infused mice vs EV-control. Disrupting ACVR1C function leads to impaired LTP independent two-sample *t* test (two-tailed): (t(10) = 3.795, P = 0.0035) relative to EV control (EV: n = 5, ACVR1C-KD: n = 5). ####P < 0.0001 compared with training day (within group). *P < 0.05, ***P < 0.001. Data are presented as mean ± SEM. Source data are provided as a Source Data file.

has no effect on reactivity to the shock on training day given similar movement during shock onset (Fig. S11B). Following behavior, we examined LTP (Fig. 5G) in dorsal hippocampal slices from a randomly selected subset of mice that were tested in OLM using the same methods as the *Acvr1c* kinase-dead construct cohorts in Fig. 4. Figure 5H summarizes the mean potentiation of LTP during the last 10 min. *Acvr1c* overexpression in area CA1 results in enhanced LTP relative to AAV-EV (Fig. 5H). *Acvr1c* overexpression of the kinase-dead and wildtype constructs was confirmed utilizing immunofluorescence to validate viral spread, quantify mean intensity of the FLAG tag and examine injection precision for CA1 region of the hippocampus (Fig. S12A–C, D, F). RT-qPCR was also conducted to validate greater *Acvr1c* mRNA abundance relative to EV control (Fig S12E, G) using a randomly selected cohort of mice from these experiments. To determine whether *Acvr1c* overexpression of the kinase-dead and wildtype constructs affects *Bdnf* levels, *Bdnf IV* mRNA was measured following viral manipulations. *Acvr1c* manipulation did not impact *Bdnf IV* mRNA levels (Fig. S13). Together (Figs. 4, 5), results identify ACVR1C as a bidirectional regulator of hippocampus-dependent memory and synaptic plasticity.

## *Acvr1c* is reduced in the aging human, mouse and 5xFAD mouse hippocampus

Given misregulation of the TGF-β pathway that occurs with age and in Alzheimer's Disease (AD) patients[31–33], we first examine whether *Acvr1c* declines with age in mouse and human hippocampus. Dorsal hippocampus was obtained from 3 and 20 mo. female and male C57BL/6 J mice and processed for RT-qPCR. We also obtained human brain transcriptome data from the publicly available Genotype-Tissue Expression (GTEx) project[56]. Human ages match life phase equivalencies in mice[57]. Figure 6A (left) shows a reduction in endogenous *Acvr1c* mRNA that occurs with age in mouse hippocampus. Sex differences were not observed, indicating that *Acvr1c* is reduced with age in both sexes. Figure 6A (right) shows a reduction in transcripts per million (TPM) with age in human hippocampus. We next assessed whether *Acvr1c* declines in the context of AD (using a 5xFAD mouse model). To examine the pattern of *Acvr1c* expression throughout the lifespan, *Acvr1c* transcripts per million (TPM) values from RNA-Seq data set obtained through the MODEL-AD consortium were analyzed from 4, 8 and 12 mo. C57BL/6 J and 5xFAD female and male mice. Figure 6H displays a decline in *Acvr1c* with age, that becomes

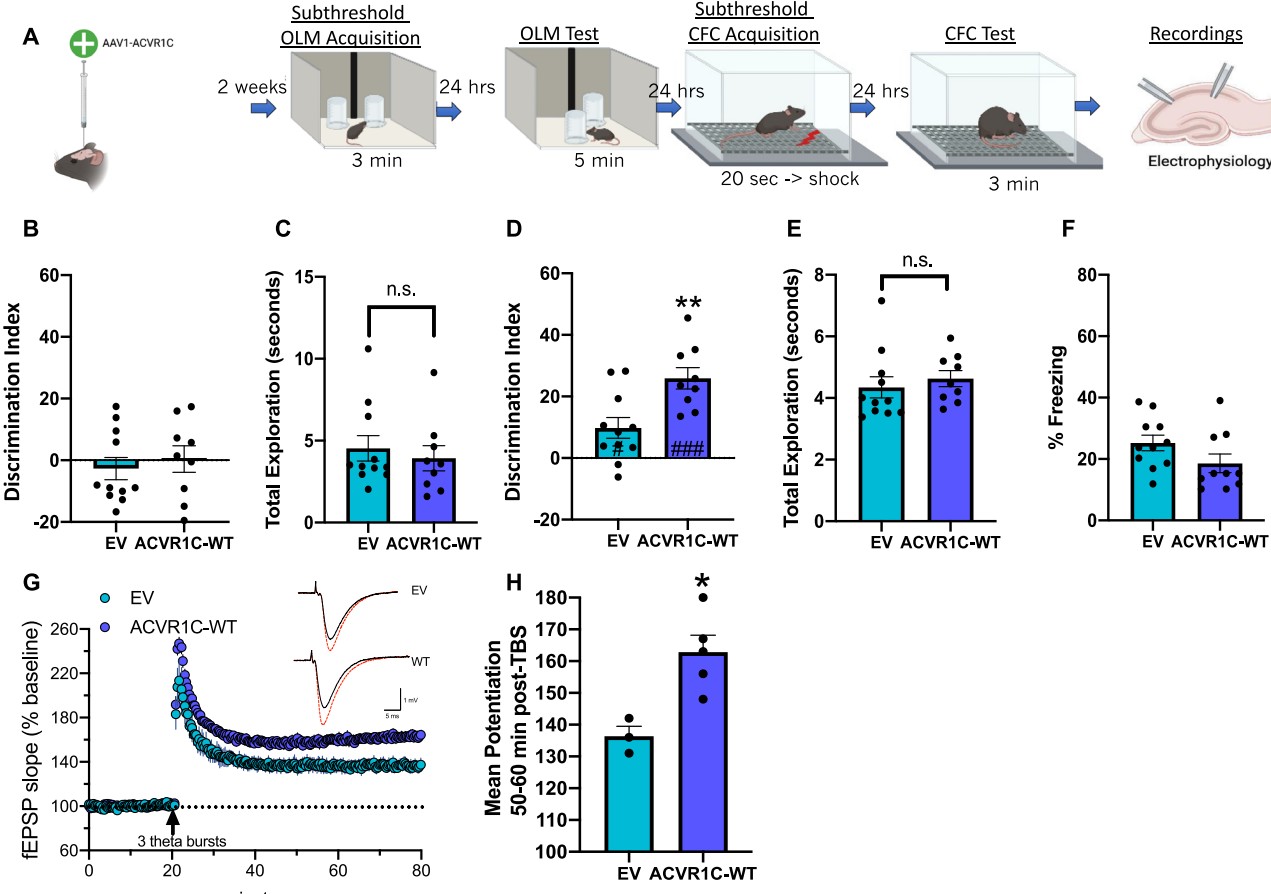

**Fig. 5 | *Acvr1c* overexpression transforms inadequate, subthreshold learning into long-term memory and facilitates LTP. A** Experimental design. Created with Biorender.com. **B** Discrimination Index (DI) scores during object location memory (OLM) training reveal no difference between groups independent two-sample *t* test (two-tailed): (t(18) = 0.564, P = 0.579), (empty vector (EV): n = 11, ACVR1C-wild type (ACVR1C-WT): n = 9). **C** Total amount of time in seconds exploring objects during training. Mice from both groups display similar levels of total object exploration independent two-sample *t*-test (two-tailed): (t(18) = 0.550, P = 0.589), (EV: n = 11, ACVR1C-WT: n = 9). **D** *Acvr1c* overexpression enhances OLM independent two-sample *t* test (two-tailed): (t(18) = 3.303, P = 0.004), (EV: n = 11, ACVR1C-WT: n = 9). **E** Exploration does not differ between groups on test independent two-sample *t*-test (two-tailed): (t(18) = 0.636, P = 0.532), (EV: n = 11, ACVR1C-WT: n = 9). **F** Percent freezing during 3-minute test session. *Acvr1c* overexpression has no effect on

freezing compared with EV control independent two-sample *t*-test (two-tailed): (t(19) = 1.696, P = 0.106), (EV: n = 11, ACVR1C-WT: n = 10). **G** Measurement of LTP as mean ± SEM excitatory postsynaptic potential (fEPSP) slope as percentage of baseline overtime (ACVR1C-WT; n = 5 mice, n = 10 slices, EV; n = 3 mice, n = 6 slices). Theta burst stimulation (TBS) applied at arrow. Inset; representative traces collected during baseline (black line) and 60 min post-TBS (red line) from ACVR1C-WT and EV group. Scale = 1 mV/5 ms. **H** Mean ± SEM level of potentiation 50–60 min post TBS showing enhanced level of LTP in slices from mice overexpressing *Acvr1c* relative to slices from EV-infused mice independent two-sample *t* test (two-tailed): (t(6) = 3.51, P = 0.0127), (EV: n = 3, ACVR1C-WT: n = 5). ###P < 0.001 compared with training day (within group). *P < 0.05, **P < 0.01. Data are presented as mean ± SEM. Source data are provided as a Source Data file.

exacerbated in 5xFAD mice as observed by a significant decrease in *Acvr1c* at 8 relative to 4 mo., only in 5xFAD mice. We do not observe sex differences indicating that *Acvr1c* is reduced in hippocampus with increasing age in 5xFAD mice of both sexes. Thus, *Acvr1c* declines in the aging and AD hippocampus during timepoints when impairments in long-term memory are observed in these models due to age[53,58,59] and AD[60,61].

### *Acvr1c* overexpression enables learning and enhances plasticity in aging mice

We next aimed to determine whether enhancing *Acvr1c* in dorsal hippocampus would enhance long-term memory formation and synaptic plasticity in aging wildtype C57Bl/6 J mice. We selected 18 mo. aging mice which typically exhibit impaired OLM performance[53,58,59] (Fig. 6B–G). Mice received infusions of AAV-ACVR1C or AAV-EV. Two weeks later, 18 mo. C57BL/6 J mice were given a 10-min subthreshold OLM acquisition session (subthreshold training for aging 18 mo. mice[59]) and long term memory was tested the next day (Fig. 6D).

ACVR1C overexpression enables long-term memory formation (Fig. 6D). Importantly, mice do not show an object preference on training (Fig. 6B, C) and exploration does not differ between groups on test (Fig. 6E). After behavior, LTP was examined in hippocampus slices from a random subset of mice tested in OLM (Fig. 6F, G), using the same methods as in Figs. 4, 5. Figure 6G summarizes the mean potentiation of LTP during the last 10 min of recording. *Acvr1c* overexpression ameliorates impairments in LTP in 18 mo. C57BL/6 J vs. EV controls (Fig. 6F, G). Thus, *Acvr1c* overexpression enables learning in aging animals under subthreshold training conditions and facilitates hippocampal LTP.

### Memory and plasticity is facilitated through *Acvr1c* overexpression

We have previously observed impairments in hippocampus-dependent memory and synaptic plasticity in 12-month-old 5xFAD mice relative to wildtype littermate controls[62]. Here, we determined whether enhancing ACVR1C in dorsal hippocampus through virus-

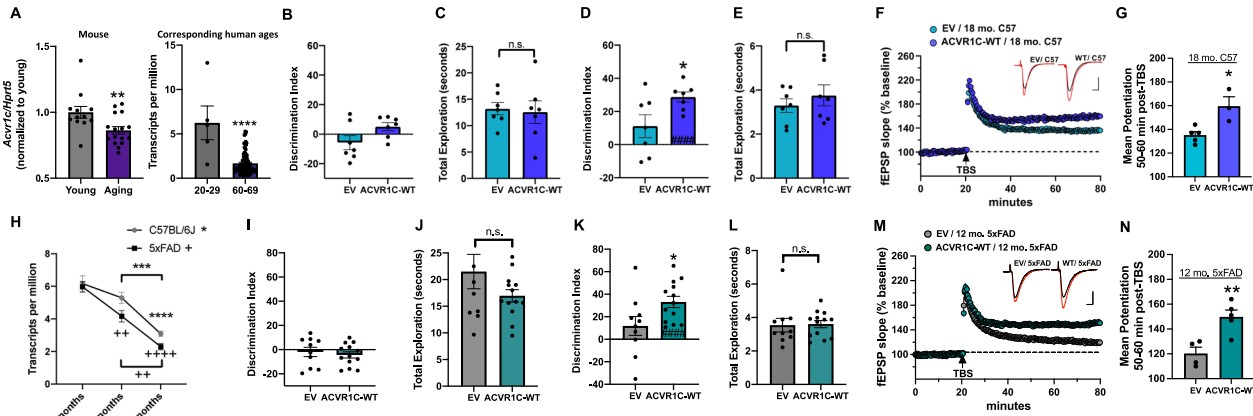

**Fig. 6 | Age and AD-related impairments in hippocampus-dependent memory and synaptic plasticity are associated with reduced *Acvr1c* expression which are ameliorated by overexpressing *Acvr1c*. A** Left: *Acvr1c* mRNA is reduced in aging mouse dorsal hippocampus independent two-sample *t*-test (two-tailed): (t(26) = 2.72, P = 0.01). Sex differences were not observed (Two-way ANOVA Sex): (F_{(1,24)} = 0.43, P = 0.50, interaction P > 0.05), indicating that *Acvr1c* is reduced with age in both sexes, (young: n = 12, aging: n = 16). Right: *Acvr1c* levels (transcripts per million (TPM)) are reduced in the aging human hippocampus independent two-sample *t*-test (two-tailed): (t(91) = 6.64, P = 0.0001). Ages displayed map to mouse ages for young (3 mo.) and aging (20 mo.), (n = 5 20–29-year-olds, n = 88 60–69-year-olds), accessed from the GTEx project expression database. **B** Discrimination Index (DI) scores during object location memory (OLM) training reveal no differences between groups independent two-sample *t* test (two-tailed): (t(12) = 1.988, P = 0.070), (n = 7/group). **C** Mice from both groups have similar levels of total object exploration independent two-sample *t* test (two-tailed): (t(12) = 0.259, P = 0.799), (n = 7/group). **D** *Acvr1c* overexpression (WT) ameliorates age-related impairments in OLM independent two-sample *t* test (two-tailed): (t(12) = 2.350, P = 0.036), (n = 7/group). **E** Exploration did not differ between groups on test independent two-sample *t* test (two-tailed): (t(12) = 0.817, P = 0.429), (n = 7/group). **F** Measurement of LTP as mean ± SEM fEPSP slope overtime (empty vector (EV); n = 5 mice, n = 10 slices; ACVR1C-WT; n = 3 mice, n = 6 slices). Theta burst stimulation (TBS) applied at arrow. Inset; representative traces collected during baseline (black line) and 60 min post-TBS (red line) from ACVR1C-WT and EV group. Scale = 1 mV/5 ms. **G** Mean ± SEM level of potentiation 50–60 min post TBS showing that *Acvr1c* overexpression enhances LTP in slices from 18 mo. C57BL6/J mice relative to EV control independent two-sample *t* test (two-tailed): (t(6) = 3.540, P = 0.0122), (EV; n = 5 mice, n = 10 slices; ACVR1C-WT; n = 3 mice, n = 6 slices). **H** RNA-Seq data displaying transcripts per million (TPM) shows decreases in *Acvr1c* with age (Three-way ANOVA), Age: (F_{(2,48)} = 54.95, P < 0.0001), that become exacerbated in 5xFAD mice (Genotype: F_{(1,48)} = 6.39, P = 0.01). Sex differences were not observed, Sex: (F_{(1,48)} = 0.40), P = 0.53, Age x Sex: P = 0.233, Age x Genotype: P = 0.387, Genotype x Sex: P = 0.037, Age x Genotype x Sex: P = 0.244. Tukey's post hoc test (compared between C57BL/6 J groups): **P < 0.01, ***P < 0.001, ****P < 0.0001, (compared between 5xFAD groups): ++P < 0.01, ++++P < 0.0001, (C57: 4 mo.: n = 10, 8 mo.: n = 11, 12 mo.: n = 10; 5xFAD: 4 mo.: n = 10, 8 mo.: n = 9, 12 mo.: n = 10). **I** Discrimination Index (DI) scores during training reveal no differences between groups independent two-sample *t* test (two-tailed): (t(21) = 0.556, P = 0.583), (EV: n = 10, ACVR1C-WT: n = 13). **J** Mice from both groups have similar levels of total object exploration on training independent two-sample *t* test (two-tailed): (t(21) = 1.459, P = 0.159), (EV: n = 10, ACVR1C-WT: n = 13). **K** *Acvr1c* overexpression (WT) ameliorates AD-related impairments in OLM independent two-sample *t*-test (two-tailed): (t(21) = 2.287, P = 0.032), (EV: n = 10, ACVR1C-WT: n = 13). **L** Exploration did not differ between groups on test independent two-sample *t*-test (two-tailed): (t(21) = 0.149, P = 0.883), (EV: n = 10, ACVR1C-WT: n = 13). **M** Measurement of LTP as mean ± SEM fEPSP slope overtime (EV; n = 4 mice, n = 8 slices; ACVR1C-WT; n = 5 mice n = 10 slices). TBS applied at arrow. Inset; representative traces collected during baseline (black line) and 60 min post-TBS (red line) from ACVR1C-WT and EV group. Scale = 1 mV/5 ms. **N** Mean ± SEM level of potentiation 50–60 min post TBS showing that *Acvr1c* overexpression enhances LTP in slices from 12 mo. 5xFAD mice relative to EV control independent two-sample *t* test (two-tailed): (t(7) = 3.852, P = 0.0063), (EV; n = 4 mice, n = 8 slices; ACVR1C-WT; n = 5 mice n = 10 slices). ###P < 0.001 compared with training day (within group). *P < 0.05, **P < 0.01, ****P < 0.0001. Data are presented as mean ± SEM.

mediated overexpression of wildtype *Acvr1c* would facilitate cognitive function in OLM and enhance LTP in 12 mo. 5xFAD males (Fig. 6I–N) (exactly as in Fig. 6B–G). *Acvr1c* overexpression enables OLM performance (Fig. 6K). Importantly, mice do not show an object preference on training (Fig. 6I–J) and exploration does not differ between groups on test (Fig. 6L). After behavior, LTP was examined in hippocampus slices from a random subset of mice tested in OLM (Fig. 6M–N), using the same methods as in Figs. 4, 5. Figure 6N summarizes the mean potentiation of LTP during the last 10 min of recording. *Acvr1c* overexpression ameliorates impairments in LTP in 12 mo. 5xFAD mice vs. EV controls (Fig. 6M, N). Lastly, we assess whether enhancing ACVR1C in dorsal hippocampus through virus-mediated overexpression of wildtype *Acvr1c* would enhance LTP in 18 mo. 5xFAD males (Fig. S14). *Acvr1c* overexpression facilitates LTP in 18 mo. 5xFAD mice vs. EV controls (Fig. S14). Taken together, results indicate that during periods in which hippocampus-dependent memory and LTP are impaired with age and in AD, *Acvr1c* is reduced in hippocampus and that conversely, enhancing ACVR1C through viral-mediated overexpression enables long-term memory formation and enhances synaptic plasticity in aging and 5xFAD mice. Therefore, ACVR1C appears to bidirectionally regulate hippocampus-dependent memory and synaptic plasticity in the adult brain and allows for long-term memory formation in addition to enhancing plasticity in both the aging, and AD brain.

## Discussion

We have previously demonstrated that cognitive benefits obtained by specific exercise patterns are maintained through sedentary delay periods which gradually decline with prolonged periods of rest, suggesting there is a molecular memory window established by specific exercise patterns. Cognitive benefits and enhancements in CA1 LTP post-exercise we observe here are congruent with our prior work in females[22] and findings of other labs[13,63,64]. Although numerous reports associate exercise with enhanced cognitive function, there is discrepancy on whether exercise increases CA1 LTP. Many studies only report enhanced LTP following exercise under aberrant conditions where LTP is already impaired including stress[65], Alzheimer's pathology[66,67], and sleep deprivation[68,69]. It is possible that the discrepancy between studies is due to variability in exercise type, length, and intensity and modality of the exercise paradigms (forced vs voluntary), including differences in strain and age. We find that molecular mechanisms engaged during the window where exercise benefits are maintained, allow for subthreshold, inadequate training

conditions (that would not lead to long-term memory) to be transformed into long-term memory. As the duration of the window extends, the effect is weakened (see Fig. 1). However, molecular mechanisms within the window appear to be maintained as a 2 day exercise (which does not facilitate plasticity or memory by itself) re-engages mechanisms that again transform subthreshold training into long-term memory. Notably, exercise benefits on hippocampal synaptic plasticity do not identically mirror behavior as benefits appear to be longer lasting in LTP overall. However, it is important to note that LTP stimulation and recordings occurred soon after behavior, where stimulation may further enhance potentiation already occurring from training. Therefore, assessing LTP in mice with the exercise conditions in the absence of training or utilization of a subthreshold stimulation parameter in the presence of training may yield results more closely aligned with behavior. Together, these results reveal how specific exercise patterns can engage a molecular memory window, but more importantly, demonstrate that the mechanisms that define that window are permissive and important for long-term memory formation/consolidation.

Thus, we applied our exercise model and RNA-sequencing approach to uncover molecular mechanisms that enable memory consolidation and may underlie maintenance of exercise benefits throughout a molecular memory window. We identify a gene coding for a type 1 receptor for the TGF-β family of signaling molecules, *Acvr1c*, as one of few genes (along with *Bdnf*) showing up-regulation in the dorsal hippocampus only under exercise conditions that drive memory consolidation under inadequate, subthreshold learning periods and enhance long term potentiation. The up-regulation of *Bdnf* mRNA aligns with the findings of Berchtold et al.[21], in which identical exercise patterns similarly elevate BDNF protein levels in hippocampus[21]. Our observation that 14-day exercise alone is sufficient for *Acvr1c* up-regulation in dorsal hippocampus, suggests that *Acvr1c* modulation occurs prior to training, suggesting that these changes may not be specific to consolidation.

To understand how *Acvr1c and Bdnf* are regulated by epigenetic mechanisms we performed several ChIP-qPCR experiments to analyze specific histone modifications that are well characterized with regard to their role in facilitating or repressing transcription. We examined H3K9Ac, H3K9Me3, H3K27Ac, and H3K27Me3. We initially hypothesized that we would observe specific patterns that correlated with the exercise effects on memory. For example, we predicted to observe H3K27 acetylation and methylation patterns and ratios that reflected the transcriptional response of *Acvr1c* and *Bdnf*, which would have been reminiscent of the histone code[70,71]. In contrast, we observed that any exercise experience leads to a reduction in the H3K27Me3/Ac ratio, suggesting that this specific lysine residue is modified after exercise and remains modified. The ratio of H3K27Me3/Ac change then may establish a local net charge[72,73] difference that allows subsequent chromatin architecture and transcriptional machinery to regulate genes like *Acvr1c* and *Bdnf* in a manner that coincides with long-term memory formation under subthreshold training conditions. This observation appeared to be specific to H3K27Me3/Ac, as a similar chromatin change through exercise was not observed with H3K9Me3/Ac. Further work will be necessary to address the open question of how exactly chromatin structure changes to facilitate this mechanism.

One potential mechanism involves self-regulated alleviation of H3K27Me3 repression at the *Acvr1c* promoter. In a study by Wang et al.[74] a reduction in H3K27Me3 repression was mediated by an ACVR1C ligand, Activin, via the Smad2-mediated reduction of the methyltransferase protein, EZH2[74]. Therefore, Activin binding to the ACVR1C receptor, followed by SMAD2 phosphorylation may mediate epigenetic repression at its own promoter and *Bdnf* by reducing methyltransferase activity. Understanding whether this mechanism mediates methyltransferase activity more broadly for this and other repressive histone modifications would have important implications

for understanding how elevated levels of *Acvr1c* through exercise or wildtype over-expression interact with the epigenome to enable learning and facilitate long-term memory. Our hypothesis suggests that exercise-induced up-regulation of *Acvr1c* involves a self-regulated alleviation of H3K27Me3 repression at the ACVR1C promoter. We further hypothesize that the binding of Activin to the ACVR1C receptor, followed by SMAD2 phosphorylation, leads to a reduction in methyltransferase activity, including EZH2, resulting in the alleviation of H3K27Me3 repression and facilitation of ACVR1C transcription. We hope to test this hypothesis in a follow up study.

To understand the role of ACVR1C in long-term memory formation in the absence of exercise, we utilized viral manipulations in the adult hippocampus to specifically disrupt and over-express *Acvr1c*. We identify ACVR1C as a bidirectional regulator of hippocampus-dependent long-term memory and synaptic plasticity. Furthermore, we find *Acvr1c* levels decrease in the hippocampus with age in C57BL6/J and 5xFAD female and male mice. We discover that *Acvr1c* over-expression enables learning and enhances synaptic plasticity in aging and 5xFAD mice. Overall, findings provide evidence for the particular TGF-β1 membrane receptor kinase, ACVR1C as a key regulator and driver of long-term memory formation and synaptic plasticity in the adult, aging and AD brain.

Our data demonstrating a reduction in *Acvr1c* in the hippocampus complement existing findings in human hippocampus where TGF-β signaling and downstream SMAD signaling become disrupted with age and in AD[28,29,31–33]. SMADs regulate transcription in the nucleus via recruitment of epigenetic machinery, including histone acetyltransferases (HATs) and DNA methyltransferases (DNMTs) to target gene promoters[23,45], which was the rationale for selecting ACVR1C to study from the RNA-Seq data. In the aging brain, memory dysfunction correlates with impairments in gene expression[75], leading several groups to hypothesize that age-related gene expression impairments involve aberrant epigenetic transcriptional repression[75]. We and others have previously demonstrated that altered histone acetylation is associated with impaired memory in aging mice[58,75–78]. Therefore, our ability to enable learning and enhance synaptic plasticity in aging and 5xFAD mice through *Acvr1c* wildtype over-expression also raises the possibility of a self-regulating mechanism where regulation downstream of ACVR1C (SMAD-dependent signaling) becomes impaired with age and AD and is maintained through self-directed aberrant epigenetic/SMAD repressor complex regulation which may repress *Acvr1c* expression. We identified that SMAD4 serves as a predicted upstream regulator for genes up regulated in exercise conditions that facilitate learning (e.g., *Acvr1c, Bdnf*) (Fig. 2K). SMAD 4 is a common mediator required for nuclear-SMAD complex insertion and transcriptional regulation[23]. Therefore, SMAD4 as a predicted upstream regulator of *Acvr1c* is consistent with the possibility of a self-regulating mechanism. Enhancing ACVR1C (expression or activity), which we find to also enable learning as with exercise in adult mice, may positively impact memory and synaptic plasticity in aging and AD by reducing epigenetic transcriptional repression via SMAD/DNMT/HDAC/HAT signaling. When the self-regulating mechanism is functioning properly, ACVR1C may play a critical role in maintaining a molecular memory window induced by an event such as exercise by priming genes (such as *Acvr1c* or *Bdnf*) or maintaining open chromatin structure via SMAD-associated DNMT/HDAC/HAT activity for re-activation upon exercise re-engagement within the molecular memory window.

Many reports indicate enhanced repressive H3K27Me3 levels in the aging brain[79], including in hippocampal tissue at the *Bdnf* promoter[80]. Understanding whether and how enhanced ACVR1C signaling and downstream SMAD-mediated signaling alleviate such repression to facilitate transcription is a critical question for future study. Furthermore, recent findings point to the ability of exercise to blunt Aβ-induced impairments in TGF-β signaling in a rat model of AD[81].

In summary, identifying and utilizing exercise patterns that enable learning under inadequate, subthreshold training sessions and exercise periods that maintain and re-activate benefits, allow for unique assessment of mechanisms which enable and drive memory consolidation and conceptually advance our understanding of mechanisms underlying exercise-facilitated cognitive function. Utilizing this exercise model, we uncover ACVR1C as a factor that gates the formation of long-term memory. We find *Acvr1c* to be up-regulated in the hippocampus when exercise drives and re-engages memory consolidation. We observe that exercise, in any amount, engages a permissive epigenetic signature at the *Acvr1c* promoter that remains engaged throughout sedentary delays. Further, we identify ACVR1C as an essential bidirectional switch of long-term memory formation where disrupted ACVR1C function under adequate learning conditions in adults impairs memory and synaptic plasticity and overexpression enables learning under inadequate training conditions in adults, aging and 5xFAD mice (in addition to enhancing synaptic plasticity). These data suggest that promoting ACVR1C through exercise or pharmacological intervention may protect against age and AD-associated cognitive impairment, providing a powerful and druggable target as a potential disease modifying treatment strategy for AD.

## Methods

### Experimental model and subject details

**Subjects.** 12-week-old C57BL/6 J male (n = 189, Figs. 1–5, 6A, S1–8, S9A, B, S10–S13), female (n = 6, Fig. 6A), 18-month-old male (n = 14, Fig. 6B–G, S9C), and 20 month-old male (n = 10 and female n = 6, Fig. 6A) C57BL/6 J mice (Jackson Laboratory) were individually housed under standard conditions (20 °C ± 1 °C; 70% ± 10% humidity; 12 h:12 h light and dark cycle) and provided *ad libitum* access to food and water. 12 (n = 23, Fig. 6I–N, S9D) and (n = 7, 18-month-old 5xFAD male mice, Fig. S14) were bred by the Transgenic Mouse Facility at UCI. 5xFAD hemizygous (B6.Cg-Tg(APPSwFlLon,PSEN1*M146L*L286V) 6799Vas/Mmjax, Stock number 34848-JAX, MMRRC) and its wildtype littermates were produced by crossing or IVF procedures with C57BL/6 J (Jackson Laboratory) females. 5xFAD mice were group housed with wildtype (C57BL/6 J) littermates to ensure accurate genotype-specific effects on behavior and synaptic plasticity. All experiments were conducted during the light phase. Experiments were conducted in accordance with the National Institutes of Health guidelines for animal care and use and were approved by the Institutional Animal Care and Use Committee of the University of California, Irvine.

### Methods details

**Exercise protocol.** We have previously developed exercise parameters that enable learning under subthreshold acquisition conditions of the OLM task in males and females and benefits can be sustained and be re-induced with brief 2-day re-introduction to exercise following a sedentary delay[11,22] (Fig. 1B). Therefore, we utilized these exercise parameters to examine effects of exercise on gene expression in the hippocampus occurring during memory consolidation, 1 h following acquisition in a subthreshold version of the object location memory task (OLM). Mice were divided into groups and individually housed in either exercise cages (equipped with running wheel) or sedentary cages (standard cage). Exercise cages are made of 9.3 in x 13.9 in x 7.7 in (length x width x height) polycarbonate and equipped with a running wheel 40 cm in circumference, 12.7 cm diameter (Lafayette Instrument). Voluntary wheel running was monitored via an 86110 Sensor/Counter plugged into a central interface connected to a dedicated PC with Scurry Activity Monitoring Software (Lafayette Instrument, model 86165). Exercise parameters consist of an initial exercise period (0 or 14d) followed by a sedentary delay (0, 7d or 14d), during which the running wheel was removed. Following this period of inactivity, some groups were assigned to receive a reactivating exercise session, consisting of 2d access to the running wheels (see Fig. 1A). Regardless of

the exercise regime, running wheels were removed the night prior to OLM training to eliminate the immediate effects of wheel running on behavior and dorsal hippocampi were rapidly dissected during consolidation, 1 h following acquisition in the 3-minute OLM task or within the same time window for exercise-only home cage mice (which did not go through OLM, (Fig. S1)). Detailed methods on tissue collection and behavior are described below.

**Object location memory task (OLM) apparatus.** The standard 10-minute and subthreshold 3-minute object-location memory (OLM) task was conducted using a set of 4 identical chambers. Each chamber was made of white-colored plastic with dimensions of 333 mm×320 mm x 310 mm (length x width x height) and contained ~1 cm deep Sani-Chips (P.J. Murphy Forest Products). A vertical matte black marking strip was adhered to one side of each chamber to serve as the spatial navigation cue. Each context was illuminated by dim yellow light (~15 lux). Exploratory behavior was recorded and was analyzed through ANY-maze tracking software version 4.99, (Stoelting Co.).

**Object-location memory task (OLM) experimental design.** OLM training, testing, and analysis were performed[11,50]. Mice were handled for 2 min per day for 5 consecutive days within the OLM training room. For mice undergoing exercise (Fig. 2), handling began on the same day for all groups regardless of where they were on their exercise regimen to ensure consistency in training day (see Fig. 2A). Following this, mice were habituated to the OLM chambers for six consecutive days. Mouse handling and habituation were overlapped for the first 2 days of habituation; habituation followed handling for each of these days. During each habituation session, mice were allowed to explore the experimental apparatus in the absence of objects for 5 minutes. Habituation sessions were analyzed (to determine the distance traveled) through ANY-maze behavioral analysis software which automatically calculated the distance moved in the experimental apparatus in meters. Reduced activity across days was used as an indicator for successful habituation (Fig. S9). Regardless of the exercise protocol, each mouse was returned to the sedentary home cage the night before OLM training to focus analysis on prior exercise impacts on cognitive function. One day after habituation, mice were trained with two identical objects (100-mL glass beakers filled with cement) in distinct locations in the habituated context using either the standard 10-minute training protocol[50] or acquisition duration adjusted to 3 min which we have previously demonstrated to be subthreshold for encoding, resulting in poor performance in both short and long-term memory[10,11,37,82]. Objects were placed 9 cm away from each other, 6 cm from the sidewalls, and 1 cm from the front wall. To assess long-term memory, mice underwent a 5 min test, 24 h post-training during which one familiar object was moved to a novel location (bottom middle) positioned 2.5 cm from the bottom wall. Exploration of the object in the novel vs. the familiar/fixed location was examined. Exploration was only scored when the mouse head pointed toward the object and came within 1 cm or when the nose touched the object. Total exploration time was recorded (t) and preference for the novel object was expressed as discrimination index (DI = ($t_{novel}$ - $t_{familiar}$) / ($t_{novel}$ + $t_{familiar}$) x 100). Positive values indicate a preference for the novel object location, whereas negative values indicate a preference for the familiar location and a score of 0 indicates absence of discrimination. For training sessions, the object designated to be moved during the test session was used as the novel object to allow training and testing DI to be directly compared. Mice that explored less than 2 s during testing or training were excluded from the study. Mice that showed a preference for either object during training (DI > ±20) were also excluded (n = 1 adult ACVR1C-WT, n = 1 18 mo. EV and n = 1 12 mo. 5xFAD EV were excluded for preference on training and n = 1 12 mo. 5xFAD EV was excluded for low exploration on test day). All training and test data collected through ANY-maze were scored manually by experimenters

blinded to the experimental groups using a scoring app to allow for precise exploration time measurements. The app only counted exploration of the object when a computer key was pressed using exploration criteria described above.

**In vitro Hippocampal slice preparation.** Shortly following OLM acquisition, mice were anesthetized with isoflurane, decapitated, and the brains were quickly removed and submerged in ice-cold, oxygenated dissection medium containing (in mM): 124 NaCl, 3 KCl, 1.25 KH$_2$PO$_4$, 5 MgSO$_4$, 0 CaCl$_2$, 26 NaHCO$_3$, and 10 glucose. Coronal hippocampal slices (340 μm) were prepared using a Leica vibrating tissue slicer (Model:VT1000S) before being transferred to an interface recording chamber containing preheated artificial cerebrospinal fluid (aCSF) of the following composition (in mM): 124 NaCl, 3 KCl, 1.25, KH$_2$PO$_4$, 1.5 MgSO$_4$, 2.5 CaCl$_2$, 26 NaHCO$_3$, and 10 glucose and maintained at 31 ± 10 C. Slices were continuously perfused with this solution at a rate of 1.75-2 ml/min while the surface of the slices were exposed to warm, humidified 95% O$_2$ / 5% CO$_2$. Recordings began after at least 2 h of incubation. Due to the limitation of being able to only run 4 mice a day, exercise and behavior were staggered for each day of recording to ensure that mice were sacrificed shortly after OLM acquisition and that slices from an exercise and control sedentary mouse were counterbalanced on 2 different extracellular rigs each day of recording.

Field excitatory postsynaptic potentials (fEPSPs) were recorded from CA1b stratum radiatum apical dendrites using a single glass pipette filled with 2 M NaCl (2-3 MΩ) in response to orthodromic stimulation (twisted nichrome wire, 65 μm diameter) of Schaffer collateral-commissural projections in CA1c stratum radiatum. Pulses were administered at 0.05 Hz using a current that elicited a 50% maximal spike-free response. After establishing a 20-minute stable baseline, long-term potentiation (LTP) was induced by delivering theta bursts, each burst consisting of four pulses at 100 Hz and the bursts themselves separated by 200 ms (i.e., theta burst stimulation or TBS). The stimulation intensity was not increased during TBS. Data were collected and digitized by NAC 2.0 Neurodata Acquisition System (Theta Burst).

The figures represent mean ± SEM. The figures illustrating the time course of LTP were produced using all slice recordings from each group, while quantification analysis of mean potentiation in the bar graphs were produced using the number of mice/group. The fEPSP slope was measured at 10–90% fall of the slope and data in figures on LTP were normalized to the last 20 min of baseline. Electrophysiological measures were analyzed using a one-way ANOVA unless otherwise specified in the text and the level of significance was set at p ≤ 0.05.

**Tissue collection for RNA-sequencing experiments.** Mice were sacrificed by cervical dislocation following completion of exercise parameters and 1-h following OLM acquisition described above. Brains were removed and dorsal hippocampi were rapidly dissected on ice and flash frozen for further processing. Tissue from all groups were taken within a 4-h window. Experiments were staggered over two separate days with equal numbers of samples/group and tissue was collected over two separate days to keep tissue collection within a short window to account for circadian effects.

**RNA-sequencing.** RNA was isolated from bilateral dorsal hippocampus using the RNeasy minikit (Qiagen, 74104). RNA quality was assessed by Bioanalyzer. Samples with an RNA integrity number >9 were included for analysis. cDNA libraries were prepared[41,47,53] using 500 ng of total RNA, according to the TruSeq RNA Sample Preparation Kit (Illumina). The mRNA was purified with poly-T oligo-attached magnetic beads and heat fragmented. The first and second strand cDNA were then synthesized and purified. The ends were blunted and the 3′ end was adenylated to prevent concatenation of the template

during adapter ligation. For each group, a unique adapter set was added to the ends of the cDNA and the libraries amplified by PCR. The quality of the library was assessed by Bioanalyzer and quantified using qRT-PCR with a standard curve prepared from a purchased sequencing library (Illumina). Samples were multiplexed with each group represented in each flow cell of the sequencer. 10 nM of each library was pooled in four multiplex libraries and sequenced on an IlluminaHiSeq 4000 instrument during a single-read 50 bp sequencing run by the Genomics High-Throughput Facility (GHTF) at the University of California, Irvine. The resulting sequencing data for each library were post-processed to produce FastQ files, then demultiplexed and filtered using both Illumina software CASAVA 1.8.2 as well as in-house software. Control reads successfully aligned to the PhiX control genome or poor-quality reads (failing Illumina's standard quality tests) were removed from analyses. The quality of the remaining sequences was further assessed using the PHRED quality scores produced in real time during the base-calling step of the sequencing run.

**Alignment to the reference genome and transcriptome.** The reads from each replicate experiment were separately aligned to the reference genome and corresponding transcriptome using the short-read aligners ELAND v2e (Illumina) and Bowtie[83]. Reads uniquely aligned by both tools to known exons or splice junctions with no more than two mismatches were included in the transcriptome. Reads uniquely aligned with more than two mismatches, or reads matching several locations in the reference genome, were removed from the analysis. The percentage of reads assigned to the reference genome and transcriptome using this protocol is reported for each group of replicates with the corresponding percentage of covered genes (Table S2).

**Gene expression and differential analysis.** Gene expression levels are directly computed from the read alignment results for each replicate. FastQ files are processed through standard Tuxedo protocol outputting FPKM values for each gene of each replicate. Differential analysis of gene expression is conducted with Cyber-T[84,85], an analysis program using Bayesian-regularized *t*-test. Comparisons were made across each pair of groups (exercise condition versus sedentary control, 1 h after acquisition) to identify genes up- or down-regulated during consolidation using the subthreshold 3-minute OLM task. For comparison, differential analysis was also conducted using linear regression models[41] (Fig. S2). Genes were first ranked by uncorrected *p* value, and only genes with both uncorrected *p* value < 0.05 were used for analysis. Fisher's exact tests (FET) were conducted to compare DEGs across conditions. The sets of genes up-regulated during consolidation in each exercise condition (compared to sedentary controls) for these 5 groups (14-0-0, 14-7-0, 14-7-2, 14-14-0, 0-0-2, see Fig. 2C; Fig S2C) were intersected to determine common up-regulated genes in exercise conditions that facilitated learning (14-0-0 and 14-7-2, Fig. 1B) to identify for further analysis. Enrichment of each group for Gene Ontology terms[86], and KEGG pathways[87,88] was assessed using DAVID[89], based on differentially expressed genes after learning. Data visualization was performed using 'matplotlib' for python and 'ggplot' for R.

**Chromatin immunoprecipitation.** Utilizing the same mice as RNA-sequencing studies, ChIP was performed on dorsal hippocampus tissue based on the protocol from the Millipore EZ-Magna ChIP kit as utilized previously[53,82,90,91]. Tissue was cross-linked with 1% formaldehyde, lysed and sonicated, and chromatin was immunoprecipitated overnight with 2μg of anti-H3K27Ac (Abcam ab4729), 10 μl of anti-H3K27Me3 (Cell Signaling #9733, clone #: C36B11), 2μg of anti-H3K9Me3 (Abcam ab176916, clone #: EPR16601) or 10μl of anti-H3k9Ac (Cell Signaling #9649 S, clone #: C5B11) or an equivalent amount of anti-rabbit IgG (negative control, Abcam ab171870). All antibodies were validated for the species and applications indicated by the manufacturer. After washing, chromatin was eluted from the beads

and reverse cross-linked in the presence of proteinase K before column purification of DNA (Qiagen, Qiaquick PCR Purification kit #28106). Primer sequences for *Acvr1c* were designed by the Primer 3 program. The following primers were used: *Acvr1c* F (5'-3') tctgtgtctgtgtctggtcc and R (5'-3') ctgaaaaggaccccaagcac. Primer sequences used for *Bdnf IV* are F (5'-3') TGCGCGGAATTCTGATTCTGG and R (5'-3') GTCCACGA-GAGGGCTCCACG and were taken from Intlekofler et al. 2013. 5 µl of input, anti-H3K27Ac, anti-H3K27Me3, anti-H3K9Ac, anti-H3K9Me3 or anti-rabbit IgG immunoprecipitate from each animal were examined in duplicate. To normalize ChIP-qPCR data, we used the percent input method. The input sample was adjusted to 100% and both the IP and IgG samples were calculated as a percent of this input using the formula $100*AE^{(adjusted\ input\ -\ Ct(IP))}$. An in-plate standard curve determined amplification efficiency (AE).

**qRT-PCR.** Quantitative RT-PCR (qRT-PCR) was performed[10,41,47,53]. RNA was isolated using RNeasy Mini Kits (Qiagen) according to the manufacturer's instructions and total RNA (50 ng) was reverse transcribed. cDNA synthesis was performed using the High-Capacity cDNA Reverse Transcription Kit (Applied Biosystems). Primers for *Acvr1c* were designed using the Roche Universal Probe Library; all primers were obtained from IDT and probes from the Roche Universal Probe Library and were used for multiplexing in the Roche Light-Cycle 480 II Machine (Roche). For *Hprt* we designed a PrimeTime qPCR assay (IDT). The following primers were used: *Acvr1c*: forward primer (5'-3'): GCCTCTGGCTCAGGCTCT, reverse primer (5'-3'): CCGACCTTTTCCTACGATTTC, UPL probe 34: AGAGGCAG; *Hprt*: forward primer (5'-3'): TGCTCGAGATGTCATGAAGG, reverse primer (5'-3'): CTTTTATGTCCCCCGTTGAC, probe: /5HEX/AT CAC ATT G/Zen/T GGC CCT CTG T/3IABkFQ/. In Fig. S12 we also designed a PrimeTime qPCR assay (IDT) for *Acvr1c* using the same primer sequences above and as in Fig. 2 with probe:/56-FAM/TGTTCTTTG/ZEN/AACCAAGAGAGGCAGACC/3IABkFQ/. All values were normalized to *Hprt* expression. Data and statistical analysis were done using Roche proprietary algorithms.

**Immunohistochemistry.** Thirty micrometer slices were collected throughout the dorsal hippocampus with a Leica CM 1850 cryostat, thaw-mounted on slides, and stored at −80 °C. Slides were fixed with 4% paraformaldehyde (15-min), rinsed in PBS (3x, 5-min), permeabilized and blocked in 0.3% Triton X-100 in 0.1 M PBS/ 5% normal goat serum (Jackson) (1 h). Slides were incubated overnight (4 °C) in rabbit antibody to Flag (1:500, Cell Signaling, ab14793, clone #: D6W5B) or in rabbit antibody to ACVR1C (1:100, LS Bio, LS-C119149-50). The following day, slides were washed and incubated for 1 h at room temperature with goat anti-rabbit Atto 488 (1:500, Sigma, ab18772) in the dark. All antibodies were validated for the species and applications indicated by the manufacturer. Slides were then washed with PBST and incubated for 15 m in 4',6'-Diamidino-2-phenylindole (DAPI) (1:10,000 in water, Invitrogen, D1306) to provide a nuclear counterstain. Slices were washed in PBS and cover slipped with VectaShield Antifade mounting medium (Vector Laboratories). Tissue was imaged by using a confocal microscope (Leica SP8). All treatment groups were represented on each slide and all images on a slide were captured with the same exposure time under nonsaturating conditions.

**AAV production.** AAV1-CMV-mAcvr1c-K222R-3flag, AAV1-CMV-mAcvr1c and AAV1-CMV-V5 were custom made at the University of California, Irvine center for Neural Circuit Mapping (https://cncm.som.uci.edu/) Viral Production Facility (titer: $3.54 \times 10^{13}$ gc/ml; $9.52 \times 10^{12}$ gc/ml). To disrupt the function of ACVR1C, a kinase-defective point mutation was created as in[92–94]; we changed the nucleotides to code for an arginine residue in place of lysine at amino acid 222.

**Surgery.** Mice were anesthetized with isoflurane (induced, 4%; maintained 2.0%) and placed in the stereotax. Injection needles were

lowered to the dorsal hippocampus at a rate of 0.2 mm/15 s (AP, −2.0 mm; ML, ± 1.5 mm, DV, −1.5 mm relative to Bregma). Two minutes after reaching the target depth, 1.0 µl of the virus carrying ACVR1C kinase dead (KD), ACVR1C wildtype over-expression or an empty vector (EV) control was infused bilaterally into the dorsal hippocampus at a rate of 6 µl/h. Injection needles remained in place for two minutes post-injection to allow the virus to diffuse. The injectors were removed at a rate of 0.1 mm per 15 s. Viral infusions were performed 2 weeks prior to behavior.

**Elevated plus maze.** In total, 24–48 h following the object location memory test, mice went through elevated plus maze testing (EPM) (Fig. S10). The elevated plus maze consists of two open arms (30 × 5 cm) and two closed arms (30 × 5 × 15 cm) that are connected by a central platform (5 × 5 cm). The maze was elevated 40 cm above the floor. During the test session, mice were recorded for 5 min on the apparatus and each mouse was placed onto the central platform facing one of the open arms. The maze was cleaned with 70% ethanol between subjects. The percentage of time spent in the open vs closed arms was scored through ANY- maze software. The elevated plus maze was conducted by an experimenter blind to the experimental groups.

**Context fear conditioning (CFC).** Contextual fear conditioning was performed[90,95]. Chambers were cleaned between each animal with 70% ethanol. Video Freeze software version 3.0.0.0, (MedAssociates) automatically scored freezing and motion index/locomotor activity. 24 h following EPM test, mice underwent context fear conditioning. During acquisition, mice were exposed to the context for either a standard 3 min period[95] followed by a 2 s (0.8 mA) foot shock or a subthreshold period of 20 s[55] followed by a 2 s (0.8 mA) foot shock and mice were removed from the context 30 s following the footshock. Long-term memory for the context-shock association was tested the following day by re-exposing mice to the same context where they received a footshock the previous day (in absence of shock) where freezing was examined (see Figs. 4F, 5F). Freezing serves as an index of fear and therefore, high freezing values indicate memory for the context-shock association.

### Quantification and statistical analysis
Sample sizes in this study were similar to those generally used in the field, including those reported in previous publications[11,41,47,48,53], although no statistical methods were used to predetermine sample sizes. Statistical analyses were performed using either one-way ANOVA (Figs. 1B, 1E, 1F, 2D, 3A–F, Figs. S1, S3 A, S7, S13), two-way ANOVA (Fig. S10A, B, Fig. S11A, B), repeated measures two-way ANOVA (Fig. S9A–D), or three-way ANOVA (Fig. 6H) followed by Tukey or Sidak- corrected *t* tests to compare individual groups or a two-tailed Student's *t* test. Simple planned comparisons to assess discrimination index (DI) scores were conducted within group to compare training and test DI using Student's *t* test (Figs. 3D, 4D, 5F). Two-way ANOVA had factors of Treatment and Day (Fig. S9A–D), Treatment and Arm (Fig. S10A, B) or Treatment and Bin (Fig. S10A, B); three-way ANOVA had factors of Age, Genotype and Sex (Fig. 6H). All statistics were performed with GraphPad Prism 8 software. Main effects and interactions for all ANOVA are described in the text. All analyses were two-tailed and required an α value of 0.05 for significance. Error bars in all figures represent SEM.

### Reporting summary
Further information on research design is available in the Nature Portfolio Reporting Summary linked to this article.

## Data availability
The raw RNA sequencing data have been deposited in NCBI's Gene Expression Omnibus (GEO) and are accessible through GEO Series

accession number GSE208615. Human brain transcriptome data were obtained from the publicly available Genotype-Tissue Expression (GTEx) project [https://www.gtexportal.org/home/downloads/adult-gtex/bulk_tissue_expression]. The paper does not report original code. Source data are provided with this paper.

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

## Acknowledgements

We wish to thank members of the Wood lab, Cotman lab and Baldi lab for scientific discussions and technical assistance. We would like to thank Dr. Eric Nestler for helpful comments on this manuscript. We would also like to thank Adeela Syed for imaging assistance and UCI's Genomic High Throughput Facility for providing their facilities and sequencing our RNA-seq libraries. This work was funded by the US National Institute on Aging (R01 AG051807 to M.A.W. and C.W.C., R21 AG067613 to M.A.W. and A.O., F32 AG071209 to A.A.K. and K99 AG078501 to A.A.K.), the US National Institute on Drug Abuse (R01 DA047981 to M.K.L. and M.A.W. and R01 DA047441 to M.A.W. and G.S.L.), the US National Institute of General Medical Sciences (R01 GM123558 to P.B.) and the US National Institute of Neurological Disorders and Stroke. Procedural images were created with Biorender.com for which we own a license to publish.

## Author contributions

A.A.K., T.N.D., C.W.B., D.P.M., N.B., P.B., C.W.C. and M.A.W. conceptualized and supervised these experiments. A.A.K. designed behavioral studies and A.A.K., T.N.D., C.W.B., J.H.B., A.S.A., A.A., Y.A., S.S. and A.R. conducted or assisted with behavioral studies. EAK performed electrophysiology experiments. D.P.M. performed ChIP-qPCR experiments. A.A.K. and J.R. performed surgeries. A.A.K. and Y.A. collected hippocampus tissue samples for RNA-seq. S.C., V.A. and P.B. performed all analysis of RNA-seq. A.A.K. performed immunohistochemistry, A.A.K, T.N.D. and J.H.B. performed qRT-PCR. A.A.K. wrote the manuscript with input from all authors.

## Competing interests

The authors declare no competing interests.
