## [Peer Review File · Nature Communications]

Specific exercise patterns generate an epigenetic molecular memory window that drives long-term memory formation and identifies ACVR1C as a bidirectional regulator of memoryREVIEWER COMMENTS

Reviewer #1 (Remarks to the Author):

The authors found that specific exercise schedules enable long-term memory formation in a weak training paradigm and that this associates with upregulation of Acvr1c and BDNF as well as as demethylation of H3K27 at these genes. Further, they provide evidence that impairing Acvr1c function prevents long-term memory formation, whereas overexpression of this gene facilitates memory formation in the absence of exercise. Memory enhancement even works in aged mice and in a model of Alzheimer's disease.

Overall this work is intriguing, but it leaves some mechanistic questions, which should be considered:

- 1) The very early enhancement of LTP suggests that specific exercise changes gene expression that impacts on the induction of LTP. However, the authors looked only for gene expression during memory consolidation and not without training in a memory task. This leaves the question whether the detected changes have occurred already before behavioural training and are not specific for consolidation. In my opinion, for the mechanistic understanding is very important to address this.
- 2) It is not clear why the authors have not studied late-LTP. There should be a better correlation with the behavior than early-LTP, in particular as Acvr1c appears to be important for memory consolidation.
- 3) Expression after transfection with the Acvr1c kinase-dead mutant has not been validated and no molecular evidence for impaired Acvr1c signaling has been provided.

Reviewer #2 (Remarks to the Author):

In this study an exercise intervention that enhances cognitive function in a subthreshold memory test (14 days of voluntary wheel running) was examined to identify the time window and underlying cellular and molecular mechanisms. Specifically, when the 14-day training period was followed by a 7-day sedentary interval and then a brief 2-day reactivating exercise session learning was enhanced but not when the sedentary interval was longer (14 days). RNA sequencing of the dorsal hippocampus identified Acvr1c as a gene of interest. The authors report that overexpression or knockdown hippocampal Acvr1c regulates memory and long-term potentiation, and that the expression of this gene is altered with aging and Alzheimer's Disease in mice and humans. The study is nice, however, the methods and experimental design are not well-described making it difficult to assess the robustness of the findings. The following comments should be considered:

1. The behavioral testing lacks a clear description of the procedure. This is particularly concerning for the Object Location Memory (OLM) test which is used extensively in this study.

The OLM protocol is described as having:

- 6 days of habituation to the apparatus,
- 1 day of presentation to the objects in the original position
- 1 day of re-introduction to objects where 1 of the objects was moved

Altogether: 8 days. However, in Figure 1A a 24 hr delay between end of the exercise paradigm and testing in OLM is shown. It is not clear how this is feasible with the 8 day timecourse, and whether the mice were sedentary during this period.

For example, assuming that animals were exposed for 14 days to running and underwent OLM with 0 days of sedentary delay and 0 days of reactivation (group 14-0-0), it is not clear whether the new location of the object, test day, occurred on "day 16" or "day 22" of the experiment. The same reasoning holds for the other experimental groups.

2. It is not clear if the groups were run simultaneously or in independent experiments. If simultaneously, then the OLM task should have been performed on the same day for all the groups. To do so, different interventions should have started at different time points to test behavior of all groups on the same day. If independently, then the groups should be compared only with the ones in which the OLM was performed on the same day. Figures 1 D, E and F the LTP data suggests that three independent experiments were performed. If this is also the case for the behavior, then OLM data should be analyzed accordingly.

3. The authors should describe if OLM data collected by use of the Any-maze software was analyzed through the program, or if data were manually analyzed by an experimenter. "All habituation, training, test, and scoring were performed by experimenters blinded to the experimental groups using a scoring app to allow for precise exploration time measurements." Please provide more details about this app, how was it used to score the behavior? It is also stated: "Habituation sessions were analyzed (to determine the distance traveled and speed) using ANY-maze behavioral analysis software. Reduced activity across days was used as an indicator for successful habituation (Fig S8)." Please clarify how the behavior was scored. Also, although it is mentioned in the text there is no speed data provided in the results or figures.

4. The OLM data throughout (Figure 1B,4B,C,5B,6B,6I) is represented as Discrimination Index (%). "Total exploration time was recorded (t) and preference for the novel object was expressed as discrimination index ($DI = (t_{\text{novel}} - t_{\text{familiar}}) / (t_{\text{novel}} + t_{\text{familiar}}) \times 100\%$). For training sessions, the object designated to be moved during the test session was used as the novel object to allow training and testing DI to be directly compared."

This formula should not be represented as percentage.

If authors prefer to use the current formula, then:

0 = absence of discrimination

positive values = preference for novel

negative values = preference for familiar

If authors prefer to present data as percentage, then use the following formula "[t Novel / (t Novel + t Familiar)] * 100", where

50% = absence of discrimination

100% > preference for novel > 50%

0 < preference for familiar < 50%

This correction is important because in the current formula, a 40% discrimination value actually indicates that the animals have a 70% preference for the novel location.

5. Figure 4. To determine whether Acvr1c regulates memory and synaptic plasticity AAV vector were used to knockdown expression and compared to AAV-EV control. However, no further validation of this intervention is provided, histological or gene expression.

6. Figure 4E. The panel shows an average of 4s of object exploration. It should be considered that this is a short amount time that may preclude robust interpretations about object discrimination, especially since the test is 5 minutes in duration. Furthermore, it is not clear why the authors do not show the total exploration time for other experiments (e.g. Figure 1).

7. In Figure 4F it is shown that the Acvr1 knockdown impairs fear learning. The methods need additional detail and clarification. In the related Figure s10 it is not clear what the three different periods depicted are. Have any habituation sessions been performed and does this represent the

"Baseline" or is it the period before the shock? It is also not clear how long the locomotion index was measured and when.

8. Figure 6. Overexpression of *Acvr1c*, please provide more than one image showing expression of the vector (Figure s7A), what was the spread of the vector throughout the hippocampus, was a systematic histological analysis performed?

9. Figure 6J,K. In graph 6J, the average total exploration value does not correspond to the individual values (dot plot) presented. Probably because not all points are shown in the 6J graph: n=7 (EV, Fig. 6J) vs. n=10 (EV, Fig. 6K).

10. The habituation data (Figure S8) should be depicted as a line graph as it a repeated measures of the same groups/animals. In the legend it is stated:

"(C) Total distance traveled during each day of OLM habituation for 18 mo. C57 mice in Figure 5. Mice from both groups habituate to the context. "

This is not what the figure panel shows, these mice do not habituate over the 6 days. The animals habituated on the second day compared to the first, but changed their behavior on subsequent days. In panel (D) the mice in the FAD experiment (12 mo) did not habituate over the days. Please clarify how the lack of habituation was addressed for these experiments.

11. The groups have different "running distances" (as shown in Figure S5). These data are for the initial 14 days of voluntary exercise for all groups. The authors do not discuss the possible reason(s) for these differences. More concerning, these mice were used for the RNA-Seq experiments. Did the authors consider that distance run could influence gene expression?

12. Even though the data show that BDNF and the gene of interest *Acvr1c* are upregulated in 14-0-0 and 14-7-2 conditions, additional experiments may be needed to address this confounding factor (differences in running distance). For the related Figure 2D please provide the n per group for the qPCR data, the relative increase in *Acvr1c* is rather small in 14-0-0 and 14-7-2. Also, please provide the running distance data for the behavioral and synaptic plasticity experiments.

13. Whole hippocampus was used for RNA-Seq. Please provide additional experiments with the spatial localization of the *Acvr1c* gene/protein in the hippocampal subfields, including area CA3 and the dentate gyrus. Why did the authors focus the synaptic plasticity (LTP), the knockdown and overexpression experiments on area CA1?

14. Long-term potentiation (LTP) is measured in slices derived from the different exercise groups and compared to control. Fourteen days of running followed by one, but not 2 weeks of sedentary housing increases area CA1 LTP. The authors do not discuss the extant literature (e.g. Dahlin et al., 2018, PMID: 30926550) that indicates that there is some controversy with regard to exercise and whether it increases area CA1 LTP. In Figure 1 how many mice and slices per animal? Data should be analyzed across mice/group, not slices/group. Was distance run recorded for these mice?

15. Figure 4G. *Acvr1* knockdown, LTP data is derived 10 slices from 5 animals. These data should be analyzed over animals not slices.

16. Figure S9. Elevated plus maze data. Use of a 2-way ANOVA assumes the data (closed vs open arm) are independent between factors which is not the case. Please modify the analysis.

17. Discussion, paragraph 2: "We identify a gene coding for a type 29 1 receptor for the TGF- β family of signaling molecules, *Acvr1c*, as one of few genes (along with 30 *Bdnf*) up-regulation of *Bdnf* mRNA aligns with the findings of Berchtold et al. (2005), 33 in which identical exercise patterns similarly elevate BDNF protein levels in hippocampus." Were hippocampal BDNF protein levels measured in the different training conditions in this study? How does overexpression or knockdown of

Acvr1c affect BDNF protein levels?

Reviewer #3 (Remarks to the Author):

This is a very interesting paper which elucidates a novel gene (ACVR1C) in dorsal hippocampus regulated by a specific exercise regimen which has been previously demonstrated to promote long-term memory formation. The authors demonstrate that this gene is required for the exercise-dependent enhancement of hippocampus-dependent LTM and hippocampal LTP. Exercise-dependent gene expression of ACVR1C is achieved via epigenetic regulation.

Further, they find that this gene is transcriptionally downregulated by aging and is likewise downregulated in a mouse model of Alzheimer's disease. Overexpression of ACVR1C in aging or AD mice reverses cognitive deficit. The conclusions drawn by the authors are, on balance, supported by the data., Just a suggestion: The authors might consider using the last line of the discussion in the Abstract, as it very clearly summarizes the impact of this manuscript.

Major Concerns:

1. I have a singular major concern which I am confident the authors can address:

The authors rely on a key piece of behavioral data reproduced in Fig. 1 from their own previous work: Butler et al., (2019). The issue is that a significant part of the current manuscript relies on categorical interpretations of those data, which may not be entirely justified. For example, on p. 4, lines 20–24, the authors describe the results from Butler et al (2019) as follows: This pattern of exercise and memory benefit suggest that a molecular memory window (Fig 1C) must exist where initial exercise benefits are maintained by mechanisms induced with initial exercise that persist for at least 7 days (but not 14 days) because the 2-day subthreshold reactivating exercise (which does not enhance memory formation by itself) leads to increased memory performance.

In Butler et al (2019) paper. the memory enhancement brought by 14 days of exercise appears to smoothly fade between 7 and 14 days of sedentary delay. In both cases, 2 days of subsequent reactivating exercise produce again a trend towards improved memory performance, but the effect is only significant after 7 days of delay. Is that sufficient grounds to say that a "memory window" persists for less than 14 days? The authors should address the concern that the lack of significance in this case does not affirm a null effect. The observed memory enhancement is clearly not a discrete switch, and all exercise-related effects are gradually reduced after prolonged periods of rest, and so it appears that the insignificant effect after 14 days may simply represent a gradual fading of all benefit brought by training, both initial and reactivated.

What further complicates the interpretation of results is that the behavioral data themselves are not replicated in the current paper. The authors instead report LTP recordings in brain slices from mice exercised according to the same protocols. However, the patterns obtained via LTP do not exactly match the behavioral data. Most notably, while the memory benefit in Butler et al (2019). fades to ~30% by 7 days and almost disappears by 14 days, the LTF benefit reported here is not significantly perturbed by 7 days. This discrepancy, in itself, is not unusual. But it leaves open the question of whether a difference exists between LTF and behavior, or rather between the Butler et al. (2019) paper and the current manuscript.

Other (less major) concerns

2.The selection of ACVR1C needs to be better described and substantiated. The logic behind the

selection is not clear.

3. Page 9, line 15 (Figure 6): The authors claim that 18 mo. C57 male mice demonstrate impairments in memory versus younger mice, but the data are not presented. This is also true for claimed impairments in hippocampal LTP in older versus young mice. These comparisons also need to be made for WT mice versus 5XFAD mice with both behavior and electrophysiology before the claims can be made that overexpression of ACVR1C is sufficient to rescue the impairment.

4. Page 11, line 7-10: The length of time which a reduction in H3K27Me3/Ac ratio needs to be established in order to draw the conclusion that this lysine residue is modified after exercise and remains modified (a persistent effect). Only two timepoints were assayed (Figure 3) and the 14-day sedentary protocol has been shown to not enhance behavioral memory. The relevance and conclusion of a "persistent effect" needs to be revisited.

Minor Concerns:

1. Page 2, line 13-15: Run-on sentence, needs editing.

2. Page 2, line 15-19: Run-on sentence, needs editing.

3. Page 3, line 10: Remove comma after "cognitive benefits"

4. Page 3, line 31-34: Run-on sentence, needs editing.

5. Page 6, line 27: Please explain more clearly why this suggests that other chromatin modifications must be involved rather than other means of differences between the LTP and the behavioral data.

6. Page 8, line 21: Revise for grammar "Together, (our) results.."

7. Page 9, line 3, and Figure 6: A better way of showing the effect in panel H would be by using a line-graph instead of histograms.

8. Figure S10: Text in figure needs to be fixed.

9. Page 10, line 6: Figure S11 does not demonstrate that there are impairments in LTP in 10 mo. 5XFAD males. These data need to be shown, though, as I've mentioned in the major concerns.

Response to Reviews:

We would like to thank the reviewers for their excellent insight, comments, and suggestions. We believe that these comments have significantly strengthened this manuscript from its original form. We have worked to address all the reviewers' comments to the best of our ability. We respond below point-by-point. Text adjustments based on reviewer comments are noted in red within the revised manuscript.

Reviewer 1: The authors found that specific exercise schedules enable long-term memory formation in a weak training paradigm and that this associates with upregulation of *Acvr1c* and BDNF as well as as demethylation of H3K27 at these genes. Further, they provide evidence that impairing *Acvr1c* function prevents long-term memory formation, whereas overexpression of this gene facilitates memory formation in the absence of exercise. Memory enhancement even works in aged mice and in a model of Alzheimer's disease.

Overall this work is intriguing, but it leaves some mechanistic questions, which should be considered:

- 1) The very early enhancement of LTP suggests that specific exercise changes gene expression that impacts on the induction of LTP. However, the authors looked only for gene expression during memory consolidation and not without training in a memory task. This leaves the question whether the detected changes have occurred already before behavioural training and are not specific for consolidation. In my opinion, for the mechanistic understanding is very important to address this.***

We thank the reviewer for this insightful comment. To address this comment, we examined whether exercise alone, in the absence of training modulates *Acvr1c* mRNA levels. We ran a separate cohort of 12-week-old C57BL/6J mice (age-matched to other mice in figure 1 and 2) through 2-week voluntary exercise (14-0-0). Here, half of the mice (n=6) received the 3-minute subthreshold object location memory acquisition session and the other half (n=6) remained in their home cage and did not go through acquisition. Running wheels were removed the night prior to OLM training to eliminate the immediate effects of wheel running on behavior. Dorsal hippocampi were rapidly dissected 1 hour following acquisition in the 3-minute OLM task or within the same time window for exercise-only home cage mice (which did not go through OLM). *Acvr1c* mRNA was assessed in dorsal hippocampus tissue and both exercise groups (training and non-training) were compared to sedentary homecage controls. *Acvr1c* mRNA was enhanced in both exercise groups, demonstrating that exercise alone, in the absence of training is sufficient to modulate *Acvr1c* mRNA levels. These data have been encoded into supplementary (see Figure S1) and data embedded into both the results section (page 6 lines 2-8) and discussion (page 12, lines 14-17). As the reviewer points out, this is very important for mechanistic understanding, as it suggests that the right parameters of exercise are sufficient to induce *Acvr1c* expression prior to sub-threshold learning. We have addressed this shift in understanding in the discussion of the revised manuscript.

- 2) It is not clear why the authors have not studied late-LTP. There should be a better correlation with the behavior than early-LTP, in particular as Acvr1c appears to be important for memory consolidation.***

Yes, the differences between early-LTP and late-LTP could be important to explore further. In our hands, using theta-burst LTP (a BDNF-dependent form of LTP) and measuring potentiation for an hour after the stimulus has been revealing as shown in Figure 1. We did

not pursue things in more depth as generating the different exercise groups and performing LTP measurements (we can handle 2 mice per day) becomes a major undertaking. The only group that did not correlate with the behavior was the 14-7-2 group. As we mention in the manuscript, theta-burst stimulation in a slice is quite different than subthreshold incidental learning (object location memory task). We were actually surprised how well the other groups mirrored each other between LTP and behavior. It is also important to note that LTP stimulation and recordings occurred soon after behavior, where stimulation may further enhance potentiation already occurring from training; we have incorporated this in the discussion. The other intriguing aspect that was beyond the scope of this project, in addition to a better exploration of e-LTP/l-LTP, is that the exercise-enhanced potentiation may become transcription-dependent much earlier than normal as perhaps eluded to by reviewer 1, comment 1. This is what we observed in our Vecsey 2007 paper using HDAC inhibition (HDAC inhibition primes chromatin and gene expression leading to transcription-dependent potentiation nearly immediately after stimulation). The increase of *Acvr1c* expression with exercise alone may not only open chromatin but also direct gene expression which opens new doors for investigation. We hope to continue the exploration of exercise effects on synaptic plasticity in future studies. We have modified the discussion to address the main question raised by the reviewer.

3) *Expression after transfection with the Acvr1c kinase-dead mutant has not been validated and no molecular evidence for impaired Acvr1c signaling has been provided.*

We apologize for not including that in the original submission. Validation of the *Acvr1c* kinase-dead mutant construct has now been embedded into the manuscript. Expression has been validated via RT-qPCR to validate greater *Acvr1c* mRNA abundance relative to EV control (Fig S8D). Additionally, we confirmed viral spread and injection precision for CA1 region of the hippocampus by utilizing immunofluorescence (Fig S12 A-B, E). We included immunofluorescence quantification for both the *Acvr1c* kinase-dead and wild-type viruses (Fig S12 D, F).

With regard to evidence for impaired *Acvr1c* signaling, we are hoping to have that work in a follow up study that will be from Dr. Ashley Keiser's lab when she starts her lab early next year at Arizona State University. There is a lot of work going into identifying the right timing, right substrate (phosphorylation target), right SMAD protein, etc. That work is part of Dr. Keiser's K99/R00, and the current plan is for that body of work to appear as a follow up study to this manuscript. We hope the reviewers understand.

Reviewer 2: In this study an exercise intervention that enhances cognitive function in a subthreshold memory test (14 days of voluntary wheel running) was examined to identify the time window and underlying cellular and molecular mechanisms. Specifically, when the 14-day training period was followed by a 7-day sedentary interval and then a brief 2-day reactivating exercise session learning was enhanced but not when the sedentary interval was longer (14 days). RNA sequencing of the dorsal hippocampus identified *Acvr1c* as a gene of interest. The authors report that overexpression or knockdown hippocampal *Acvr1c* regulates memory and long-term potentiation, and that the expression of this gene is altered with aging and Alzheimer's Disease in mice and humans. The study is nice, however, the methods and experimental design are not well-described making it difficult to assess the robustness of the findings. The following comments should be considered:

1) *The behavioral testing lacks a clear description of the procedure. This is particularly concerning for the Object Location Memory (OLM) test which is used*

extensively in this study.

The OLM protocol is described as having:

- 6 days of habituation to the apparatus,*
- 1 day of presentation to the objects in the original position*
- 1 day of re-introduction to objects where 1 of the objects was moved*

Altogether: 8 days. However, in Figure 1A a 24 hr delay between end of the exercise paradigm and testing in OLM is shown. It is not clear how this is feasible with the 8 day timecourse, and whether the mice were sedentary during this period.

For example, assuming that animals were exposed for 14 days to running and underwent OLM with 0 days of sedentary delay and 0 days of reactivation (group 14-0-0), it is not clear whether the new location of the object, test day, occurred on "day 16" or "day 22" of the experiment. The same reasoning holds for the other experimental groups.

We apologize for the confusion with the experimental approach and thank the reviewer for the detailed comment so we can understand where we made assumptions. We have edited the methods section to clearly describe the object location memory procedure and timeline (see page 20, lines 18-29). Although we mention that Figure 1B is adapted from Butler et al., 2019, this was not stated for Figure 1A. We have now included this as well as a note with this citation for detailed methods. We also provided references for past studies in which we have used the 3 min and 10 min subthreshold and threshold protocols.

To depict the timeline of the exercise periods, sedentary periods, and behavioral procedures more clearly, we have created a new schematic for Figure 2. This schematic displays individual groups to indicate the timeline for such events and is identical to the order of procedures in Butler et al., 2019 (see Figure 2A). Regardless of the exercise protocol, each mouse was returned to the sedentary home cage the night before OLM training to focus analysis on prior exercise impacts on cognitive function. Therefore, we have edited Figure 1A and 2B to indicate that OLM occurred the next day following reactivating exercise rather than 24 hours later.

2) *It is not clear if the groups were run simultaneously or in independent experiments.*

If simultaneously, then the OLM task should have been performed on the same day for all the groups. To do so, different interventions should have started at different time points to test behavior of all groups on the same day.

Figures 1 D, E and F the LTP data suggests that three independent experiments were performed. If this is also the case for the behavior, then OLM data should be analyzed accordingly.

This is a good point and prompted us to make the schematic indicating the exercise and behavioral timeline for each group (see Figure 2A). Mice from all groups were handled, habituated, trained, and tested on the same days for all RNA-Seq and behavioral experiments. Due to this, different exercise interventions started at different time points as reviewer notes.

Due to the limitation of being able to only run 2 mice a day for LTP experiments, exercise and behavior were staggered for each day of recording to ensure that mice were sacrificed shortly after OLM acquisition and control, sedentary mice were included each day; we have included these details in the LTP methods section "In vitro Hippocampal Slice Preparation".

Figures 1D, E and F were indeed run in separate cohorts. Due to this, we have also included the stats for each individual graph in addition to the one-way ANOVA with all groups in figure 1G.

- 3) ***The authors should describe if OLM data collected by use of the Any-maze software was analyzed through the program, or if data were manually analyzed by an experimenter.***

"All habituation, training, test, and scoring were performed by experimenters blinded to the experimental groups using a scoring app to allow for precise exploration time measurements." Please provide more details about this app, how was it used to score the behavior? It is also stated: "Habituation sessions were analyzed (to determine the distance traveled and speed) using ANY-maze behavioral analysis software. Reduced activity across days was used as an indicator for successful habituation (Fig S8)."

Please clarify behavior how the was scored. Also, although it is mentioned in the text there is no speed data provided in the results or figures.

We have now incorporated scoring experimental details into the methods section (see page 21, lines 12-15). As only distance traveled was analyzed through the ANY-maze software, we have removed "speed" this from the text. We thank the reviewer for catching this.

- 4) ***The OLM data throughout (Figure 1B,4B,C,5B,6B,6I) is represented as Discrimination Index (%). "Total exploration time was recorded (t) and preference for the novel object was expressed as discrimination index ($DI = (t_{\text{novel}} - t_{\text{familiar}}) / (t_{\text{novel}} + t_{\text{familiar}}) \times 100\%$). For training sessions, the object designated to be moved during the test session was used as the novel object to allow training and testing DI to be directly compared."***

This formula should not be represented as percentage.

If authors prefer to use the current formula, then:

0 = absence of discrimination

positive values = preference for novel

negative values = preference for familiar

If authors prefer to present data as percentage, then use the following formula "[t Novel / (t Novel + t Familiar)] * 100", where

50% = absence of discrimination

100% > preference for novel > 50%

0 < preference for familiar < 50%

This correction is important because in the current formula, a 40% discrimination value actually indicates that the animals have a 70% preference for the novel location.

We thank the reviewer for catching this issue. We have edited the methods section to state that positive values indicate a preference for the novel object location, whereas negative values indicate a preference for the familiar location and a score of 0 indicates absence of discrimination (see page 21, lines 6-8). The % for the discrimination index scores has also been removed from all figures and text.

5) Figure 4. To determine whether *Acvr1c* regulates memory and synaptic plasticity AAV vector were used to knockdown expression and compared to AAV-EV control. However, no further validation of this intervention is provided, histological or gene expression.

Please see response to Reviewer 1, comment 3 above as it relates to similar comment. Validation of the *Acvr1c* kinase-dead mutant construct has now been embedded into the manuscript. Expression has been validated via RT-qPCR to validate greater *Acvr1c* mRNA abundance relative to EV control (Fig S12D).

6) Figure 4E. The panel shows an average of 4s of object exploration. It should be considered that this is a short amount time that may preclude robust interpretations about object discrimination, especially since the test is 5 minutes in duration. Furthermore, it is not clear why the authors do not show the total exploration time for other experiments (e.g. Figure 1).

Thank you for this comment. We have now made clear in the legend that OLM data in Figure 1 is adapted from Butler et al. (2019) and thus the exploration time is in that paper. We have found that total exploration time depends on the device used to score the OLM data, although the DI scores do not differ by scoring device. For example, stop watches result in higher overall exploration times, whereas the scoring app used here, we observe lower overall exploration times. Overall exploration times on OLM test are similar to our previous studies where this app was also used to score object exploration (Dong et al., 2022, Keiser et al., 2021, Kwapis et al., 2019, Shu et al., 2018, Kwapis et al., 2018, Vogel-Ciernia et al., 2015). Longer exploration with stop watches is largely due to the requirement to start *and* stop the watch during each exploration bout. The scoring app that we utilize here does not require a button to be pressed to stop the scoring. Rather, scoring is only counted when the button is actively being pressed, allowing for more precise exploration times that remain consistent with DI scores when using stop watches. Importantly, we avoid scoring mice with too low exploration times by excluding mice from the study if exploration is less than 2 seconds during training or testing. Mice that showed a preference for either object during training ($DI > \pm 20$) were also excluded. Overall, we have demonstrated rigor and reproducibility using this task and the methods to collect and interpret the data across numerous studies, several labs, adult mice and aging mice, and many different genetic and viral manipulations.

7) In Figure 4F it is shown that the *Acvr1* knockdown impairs fear learning. The methods need additional detail and clarification. In the related Figure s10 it is not clear what the three different periods depicted are. Have any habituation sessions been performed and does this represent the "Baseline" or is it the period before the shock? It is also not clear how long the locomotion index was measured and when.

We have made edits to the "Context Fear Conditioning" methods section where we have added additional detail and made edits for clarification (see page 26, lines 4-12). When mentioning shock reactivity in the results section, we also specified that the shock was occurring on training day. For the supplemental shock reactivity figure, we have included descriptions in the legends that depict what "Baseline", "Shock" and "Post-shock" refers to, and when the motion index is being measured. "Baseline" refers to the motion occurring before the shock, "Shock" refers to the motion during the 2 second footshock, and "Post-shock" refers to the motion during the 30 seconds following the shock. How the motion index is calculated is now specified in methods.

8) Figure 6. Overexpression of *Acvr1c*, please provide more than one image showing expression of the vector (Figure S7A), what was the spread of the vector throughout the hippocampus, was a systematic histological analysis performed?

We have now added more images and confirmed viral spread and injection precision for CA1 region of the hippocampus by utilizing immunofluorescence (Fig S12 A-B, E). We have now included immunofluorescence quantification for both the *Acvr1c* kinase-dead and wild-type viruses (Fig S12 D, F).

9) Figure 6J,K. In graph 6J, the average total exploration value does not correspond to the individual values (dot plot) presented. Probably because not all points are shown in the 6J graph: n=7 (EV, Fig. 6J) vs. n=10 (EV, Fig. 6K).

We thank the reviewer for this comment which helped us to clarify. Figure 6J is referring to the training data and corresponds with Figure 6I, whereas Figure 6L is referring to the test data and corresponds to Figure 6K. We have now specified that Figure 6J refers to total object exploration on training day in the legend. We have also checked that the same data points within each EV or *Acvr1c*-WT group are displayed on graphs 6I-L as well as B-E.

10) The habituation data (Figure S8) should be depicted as a line graph as it a repeated measures of the same groups/animals. In the legend it is stated: “(C) Total distance traveled during each day of OLM habituation for 18 mo. C57 mice in Figure 5. Mice from both groups habituate to the context. “ This is not what the figure panel shows, these mice do not habituate over the 6 days. The animals habituated on the second day compared to the first, but changed their behavior on subsequent days. In panel (D) the mice in the FAD experiment (12 mo) did not habituate over the days. Please clarify how the lack of habituation was addressed for these experiments.

We have now updated all 4 habituation graphs for Figure S9 (note updated figure number), and data are now depicted as a line graph. We have also updated the legend for panel C in Figure S9 to indicate which days of habituation result in lower distance travelled relative to day 1. “Mice from both groups habituate to the context as indicated by lower exploration on day 2 relative to day 1 for the EV group and lower exploration on day 2, 4 and 5 in the ACVR1C-WT mice.” Deficits in locomotion are common in both aging (e.g. Boyer et al., 2019, Bordner et al., 2011, Barreto et al., 2010, Singhal et al., 2020) and 5xFAD (O’Leary et al., 2018, O’Leary et al., 2020, Smith & Hopp, 2023, Oblak et al., 2021, Forner et al., 2021) mice as observed by lower overall distance travelled as compared to younger cohorts. As expected, aging and 12 mo. 5xFAD groups exhibit lower distance traveled on the first day of habituation compared with the younger cohorts in our study. Due to this, it was not unexpected that distance travelled did not decrease for each habituation day 2-6 as the first day of habituation for the aging and 5xFAD groups was similar in level to the lowest level of exploration on day 6 of habituation for the younger cohorts, indicating a potential floor effect. Although our measurements cannot pick this up, habituation to the context may still be taking place. Intact context exploratory behavior (no difference in time spent near the walls vs the center of the cage by age) has been observed in studies showing a decline in locomotion with age (Boyer et al., 2019).

11) The groups have different "running distances" (as shown in Figure S5). These data are for the initial 14 days of voluntary exercise for all groups. The authors do

not discuss the possible reason(s) for these differences. More concerning, these mice were used for the RNA-Seq experiments. Did the authors consider that distance run could influence gene expression?

We appreciate the reviewer's comment which prompted us to assess the relationship between running distance within the initial 2-week period and gene expression. We included an additional panel in Figure S3 (panel B) (note updated figure number) where we performed a simple linear regression analysis to test if the initial 2-week running distance predicted relative *Acvr1c/Hprt* levels. The overall regression analysis did not yield statistically significant results ($R^2 = 0.093$, $F(1,31) = 3.209$, $p = 0.083$), suggesting that the model, which incorporated running distance as the independent variable, did not achieve statistical significance. The R-squared value of 0.093 indicates that running distance explains only 9.3% of the variation in relative *Acvr1c/Hprt* mRNA levels. This relatively low R-squared value underscores the modest association between running distance and *Acvr1c* expression levels, further supporting the notion that the relationship is not strongly established based on the current model. We employ voluntary running here to remove the stress component of forced running, which could impact the OLM component. It is also worth noting that although initial 14-day running distance varied across the groups, *Acvr1c* expression in 14-0-0 and 14-14-0 varied considerably despite similar initial 14-day exercise. Diverse *Acvr1c* expression profiles are also evident in 14-7-0 and 14-7-2 whose running distances were closely aligned. Regardless, we appreciate the reviewers' comment and have added this as a caveat to the interpretation to the overall RNA-Seq results.

12) Even though the data show that BDNF and the gene of interest Acvr1c are upregulated in 14-0-0 and 14-7-2 conditions, additional experiments may be needed to address this confounding factor (differences in running distance). For the related Figure 2D please provide the n per group for the qPCR data, the relative increase in Acvr1c is rather small in 14-0-0 and 14-7-2. Also, please provide the running distance data for the behavioral and synaptic plasticity experiments.

Thank you for this comment. Please see point 11 above where we included an additional panel in Figure S3B where we performed a simple linear regression analysis to better understand the relationship between running distance and *Acvr1c* expression. We have now updated the legend for Figure 2 to include n's for RT-qPCR data. Although the qPCR data reveal a relatively small increase in *Acvr1c* expression, RNA-Seq being a more sensitive and high-throughput method can provide a more nuanced assessment of gene expression dynamics, allowing for detection of more modest changes that may be overlooked by RT-qPCR. Running distance data for behavior in Figure 1B can be found in the Butler et al., 2019 paper and we have updated the legend to cite this paper at the beginning to ensure clarity for future readers.

13) Whole hippocampus was used for RNA-Seq. Please provide additional experiments with the spatial localization of the Acvr1c gene/protein in the hippocampal subfields, including area CA3 and the dentate gyrus. Why did the authors focus the synaptic plasticity (LTP), the knockdown and overexpression experiments on area CA1?

Dorsal hippocampus was used for RNA-Seq experiments. Although dorsal hippocampus is stated throughout the manuscript, a couple of sentences in the discussion did not specify. We have edited this so that it now says dorsal. "Whole dorsal hippocampus" was also stated in results, and we have now removed "whole" to avoid confusion. We have now added additional

immunohistochemistry experiments to display the spatial localization of ACVR1C which we observe to be abundant throughout the hippocampus. This high-resolution figure displaying an overview of hippocampus and a high magnification image displaying CA1 can be found in supplementary (see figure S8). The object location memory task requires the dorsal hippocampus (Barrett et al., 2011, McQuown et al., 2011) which is where we measured hippocampal CA1 LTP and conducted viral manipulations. Thus, the physiology aligns very well with the behavior, as the Wood lab has demonstrated in several studies (Keiser et al., 2021; Dong et al., 2022; Kwapis et al., 2018; Shu et al., 2018; Barrett et al., 2011; Lopez et al., 2016; Ciernia et al., 2017).

14) Long-term potentiation (LTP) is measured in slices derived from the different exercise groups and compared to control. Fourteen days of running followed by one, but not 2 weeks of sedentary housing increases area CA1 LTP. The authors do not discuss the extant literature (e.g. Dahlin et al., 2018, PMID: 30926550) that indicates that there is some controversy with regard to exercise and whether it increases area CA1 LTP. In Figure 1 how many mice and slices per animal? Data should be analyzed across mice/group, not slices/group. Was distance run recorded for these mice?

In our previous study incorporating CA1 LTP post-exercise in females (Dong et al., 2022), we discuss work, at length, that is consistent with enhanced LTP following exercise and work which reports opposing effects, including discussing the study above that the reviewer notes. Although we cannot discuss these discrepancies in as much depth here due to space restrictions, we have incorporated these important studies as well as potential rationale for differences in CA1 LTP results post exercise in the first paragraph of the discussion. “Cognitive benefits and enhancements in CA1 LTP post-exercise we observe here are congruent with our prior work in females²² and findings of other labs^{13,63,64}. Although numerous reports associate exercise with enhanced cognitive function, there is discrepancy on whether exercise increases CA1 LTP. Many studies only report enhanced LTP following exercise under aberrant conditions where LTP is already impaired including stress⁶⁵, Alzheimer’s pathology^{66,67}, and sleep deprivation^{68,69}. It is possible that the discrepancy between studies is due to variability in exercise type, length, and intensity and modality of the exercise paradigms (forced vs voluntary), including differences in strain and age.” For instance, many of these studies use forced exercise (Dao et al., 2013, 2015; D’Arcangelo et al., 2017; Saadati et al., 2015; Tsai et al., 2018; Zagaar et al., 2012, 2013) with varied protocols, while only a few studies employ voluntary wheel running (Ivy et al., 2020; Saadati et al., 2014) with different exercise duration. We also state the following as a potential explanation for enhanced LTP post-exercise in area CA1. “However, it is important to note that LTP stimulation and recordings occurred soon after behavior, where stimulation may further enhance potentiation already occurring from training”. We have now included the number of mice and slices per animal in the legend for Figure 1. For all LTP figures (Figure 1, 4, 5, 6 and S14) we now report the data analyzed across mice/group. We have also included individual data points for each graph. Unfortunately, running distance was not collected for LTP studies.

15) Figure 4G. Acrv1 knockdown, LTP data is derived 10 slices from 5 animals. These data should be analyzed over animals not slices.

For all LTP figures (Figure 1, 4, 5, 6 and S14) we now report the data analyzed across mice/group.

16) Figure S9. Elevated plus maze data. Use of a 2-way ANOVA assumes the data (closed vs open arm) are independent between factors which is not the case. Please modify the analysis.

We thank the reviewer for catching this. Analyses were double checked and indeed, repeated measures two-way ANOVA was conducted on the EPM data. We have now fixed the text in the figure legend so that it reflects this type of analysis.

17) Discussion, paragraph 2: “We identify a gene coding for a type 29 1 receptor for the TGF- β family of signaling molecules, *Acvr1c*, as one of few genes (along with 30 *Bdnf*) ... up-regulation of *Bdnf* mRNA aligns with the findings of Berchtold et al. (2005), 33 in which identical exercise patterns similarly elevate BDNF protein levels in hippocampus.” Were hippocampal BDNF protein levels measured in the different training conditions in this study? How does overexpression or knockdown of *Acvr1c* affect BDNF protein levels?

While our RNA-sequencing results mirror a similar pattern to the Berchtold et al., 2005 study, BDNF protein levels were not measured here. We did conduct an additional experiment where we measured *Bdnf4* mRNA in mice following viral manipulations and included this data as supplementary Figure S13. We find that ACVR1C manipulation does not impact *Bdnf4* mRNA levels.

Reviewer 3: This is a very interesting paper which elucidates a novel gene (ACVR1C) in dorsal hippocampus regulated by a specific exercise regimen which has been previously demonstrated to promote long-term memory formation. The authors demonstrate that this gene is required for the exercise-dependent enhancement of hippocampus-dependent LTM and hippocampal LTP. Exercise-dependent gene expression of ACVR1C is achieved via epigenetic regulation.

Further, they find that this gene is transcriptionally downregulated by aging and is likewise downregulated in a mouse model of Alzheimer’s disease. Overexpression of ACVR1C in aging or AD mice reverses cognitive deficit. The conclusions drawn by the authors are, on balance, supported by the data. **Just a suggestion: The authors might consider using the last line of the discussion in the Abstract, as it very clearly summarizes the impact of this manuscript.**

We thank the reviewer for this suggestion which we have implemented in the updated abstract.

1) I have a singular major concern which I am confident the authors can address:

The authors rely on a key piece of behavioral data reproduced in Fig. 1 from their own previous work: Butler et al., (2019). The issue is that a significant part of the current manuscript relies on categorical interpretations of those data, which may not be entirely justified. For example, on p. 4, lines 20–24, the authors describe the results from Butler et al (2019) as follows: This pattern of exercise and memory benefit suggest that a molecular memory window (Fig 1C) must exist where initial exercise benefits are maintained by mechanisms induced with initial exercise that persist for at least 7 days (but not 14 days) because the 2-day subthreshold reactivating exercise (which does not enhance memory formation by itself) leads to increased memory performance.

In Butler et al (2019) paper. the memory enhancement brought by 14 days of exercise appears to smoothly fade between 7 and 14 days of sedentary delay. In both cases, 2 days of subsequent reactivating exercise produce again a trend towards improved memory performance, but the effect is only significant after 7 days of delay. Is that sufficient grounds to say that a “memory window” persists for less than 14 days? The authors should address the concern that the lack of significance in this case does not affirm a null effect. The observed memory enhancement is clearly not a discrete switch, and all exercise-related effects are gradually reduced after prolonged periods of rest, and so it appears that the insignificant effect after 14 days may simply represent a gradual fading of all benefit brought by training, both initial and reactivated.

What further complicates the interpretation of results is that the behavioral data themselves are not replicated in the current paper. The authors instead report LTP recordings in brain slices from mice exercised according to the same protocols. However, the patterns obtained via LTP do not exactly match the behavioral data. Most notably, while the memory benefit in Butler et al (2019). fades to ~30% by 7 days and almost disappears by 14 days, the LTF benefit reported here is not significantly perturbed by 7 days. This discrepancy, in itself, is not unusual. But it leaves open the question of whether a difference exists between LTF and behavior, or rather between the Butler et al. (2019) paper and the current manuscript.

We absolutely agree with the reviewer and appreciate these comments. We have revised the manuscript to reflect that the lack of significance of 2 days of reactivating exercise following a 14-day sedentary delay does not affirm a null effect. The revised manuscript also now presents the alternative interpretation that there is gradual decline in memory enhancement with prolonged periods of rest in the abstract, results section and discussion.

In addition to the Butler et al., 2019 paper which we reference, we have also replicated the behavioral and electrophysiology findings in female animals in our recent paper (Dong et al., 2022) applying the same methods used here. We have incorporated these data in the first paragraph of the discussion section to further strengthen the basis of the work conducted here and illustrate the rigor and reproducibility of the exercise effect on memory formation. Still, we agree with the reviewer that the memory enhancement we observe here with reactivating exercise does not reflect a discrete switch and have adjusted wording throughout accordingly. Regarding the comment that LTP data do not match the behavior exactly, we have incorporated discussion on this point in the first paragraph of the discussion section of the manuscript. “Notably, exercise benefits on hippocampal synaptic plasticity do not identically mirror behavior as benefits appear to be longer lasting in LTP overall. However, it is important to note that LTP stimulation and recordings occurred soon after behavior, where stimulation may further enhance potentiation already occurring from training. Therefore, assessing LTP in mice with the exercise conditions in the absence of training or utilization of a subthreshold stimulation parameter in the presence of training may yield results more closely aligned with behavior.”

2) The selection of ACVR1C needs to be better described and substantiated. The logic behind the selection is not clear.

We appreciate this reviewer’s suggestion. We have provided additional rationale for pursuing ACVR1C. These changes can be found in the third paragraph of the introduction. “Therefore, the selection of ACVR1C for further assessment of its role in hippocampus-dependent memory and

synaptic plasticity is motivated by its transcriptional regulatory role through SMAD-mediated signaling. Additionally, the inherent druggability and literature supporting a role for type I receptors in cognitive function and synaptic plasticity poise ACVR1C as a promising target. Lastly, because the SMAD proteins interact with chromatin modifying enzymes, like CBP, it may be that ACVR1C may participate in the signaling and mechanisms that ultimately modify the epigenome in response to exercise making it a desirable target from our perspective. Edits have also been made throughout the text to clarify rationale for selection of ACVR1C. We appreciate the reviewer highlighting this.

- 3) Page 9, line 15 (Figure 6): The authors claim that 18 mo. C57 male mice demonstrate impairments in memory versus younger mice, but the data are not presented. This is also true for claimed impairments in hippocampal LTP in older versus young mice. These comparisons also need to be made for WT mice versus 5XFAD mice with both behavior and electrophysiology before the claims can be made that overexpression of ACVR1C is sufficient to rescue the impairment.**

Indeed, we apologize for the mistake here. We were referring to several of our past studies and did not reference things correctly or discuss them correctly here. We have added the correct references as well when discussing past comparisons between adult and aging mice, as well as aging wildtype and aging 5xFAD mice. We have adjusted wording throughout the manuscript for claims made regarding impairments in memory in older vs younger mice. We have also adjusted these statements for 5xFAD mice, as wildtype mice were not run alongside the 12 or 18 mo. 5xFAD mice. We now refer the effects of *Acvr1c* overexpression in aging or 5xFAD mice as “enabling learning” and “enhancing long term potentiation”, depending on the control groups used in the experiments.

- 4) Page 11, line 7-10: The length of time which a reduction in H3K27Me3/Ac ratio needs to be established in order to draw the conclusion that this lysine residue is modified after exercise and remains modified (a persistent effect). Only two timepoints were assayed (Figure 3) and the 14-day sedentary protocol has been shown to not enhance behavioral memory. The relevance and conclusion of a “persistent effect” needs to be revisited.**

The reviewer makes a great point here and we have edited the discussion as well as all sections throughout the manuscript to remove language stating a persistent effect. We are continuing to examine the epigenetic nature of what may be an ‘epigenetic molecular memory’ as well as ‘persistent’ effects, but with the data we have currently we have revised the statements to more accurately reflect the data.

- 5) 1. Page 2, line 13-15: Run-on sentence, needs editing.
2. Page 2, line 15-19: Run-on sentence, needs editing.
3. Page 3, line 10: Remove comma after “cognitive benefits”
4. Page 3, line 31-34: Run-on sentence, needs editing.
5. Page 6, line 27: Please explain more clearly why this suggests that other chromatin modifications must be involved rather than other means of differences between the LTP and the behavioral data.
6. Page 8, line 21: Revise for grammar “Together, (our) results..”
7. Page 9, line 3, and Figure 6: A better way of showing the effect in panel H would be by using a line-graph instead of histograms.
8. Figure S10: Text in figure needs to be fixed.
9. Page 10, line 6: Figure S11 does not demonstrate that there are impairments in**

LTP in 10 mo. 5XFAD males. These data need to be shown, though, as I've mentioned in the major concerns.

We thank the reviewer for catching the above points and have made the appropriate edits.

REVIEWER COMMENTS

Reviewer #1 (Remarks to the Author):

The authors have addressed my comments. Congratulations on an excellent manuscript.

Reviewer #2 (Remarks to the Author):

The authors have improved the manuscript, however, there are remaining concerns that should be addressed:

1. Re reviewer comment 4. The authors have adjusted the discrimination index (D.I.) measurement removing the percentage scores as suggested by the reviewer. However, object location memory (OLM) data does not appear to be consistent across experiments. In Fig. 4D in the AAV control group (EV), in the absence of exercise the D.I. = ~30, suggesting improved OLM. And then in Fig. 5D the exact same treatment, absence of exercise in AAV Control (EV) results in D.I. = ~8 suggesting no effect on OLM. Please explain how the same treatment and testing can result in such vastly different outcomes.

2. Re reviewer comment 5. To determine whether Acvr1c regulates memory and synaptic plasticity AAV vectors were used to knockdown expression and compared to AAV-EV control (Fig. 4). Further validation of this intervention is provided (Fig. S12) using immunocytochemistry and Acvr1c kinase-dead mutant construct and via RT-qPCR. However, it would have been better if each animal that underwent behavioral or electrophysiological testing following vector treatment had undergone histological validation for injection location, spread of vector expression across the extent of the dorsal hippocampus, as such parameters could affect the outcome of behavioral testing.

3. Re reviewer comment 6. Figure 4E: The panel shows an average of 4s of object exploration. It should be considered that this is a short amount time that may preclude robust interpretations about object discrimination, especially since the test is 5 minutes in duration. Furthermore, it is not clear why the authors do not show the total exploration time for other experiments (e.g. Figure 1).

-“Author response: Thank you for this comment. We have now made clear in the legend that OLM data in Figure 1 is adapted from Butler et al. (2019) and thus the exploration time is in that paper. ”

- Figure 1B in the manuscript represents the findings from Butler et al., 2019, essentially republishing these data to form the premise for a new paper. The reviewer is now referred to that paper to search for the total exploration times and running distances for each behavioral condition.

- The republishing of previous data that was collected with different experimenters at a different point in time and using it as the basis for a new study with the same parameters is very questionable. Indeed, while the same strain and gender of mice was used, the mice in the Butler et al. 2019 study were a different age, 7 weeks old at the time of the start of the exercise intervention and behavioral testing (in Dong et al., 2022, female mice, 8 weeks were utilized), whereas in the current study the adult mice are being tested at 12 weeks of age. Therefore, it is not clear whether the same outcomes could be expected. This experiment should be redone accordingly.

- Furthermore, this study includes aging and Alzheimer’s Disease mice. Is OLM affected by aging (e.g. Muria et al., 2007 PMID: 17049363)? Do the same exercise parameters (14-0-0, 14-7-2) modify OLM in the 18-month-old WT and 12-month-old FAD mice? For the adult mice the authors suggest that the effect of ACVR1C-WT vector treatment in 12-week-old mice on D.I. (~25, Figure 5D) is about equal to the effect of exercise in the Butler et al., 2019, in 7-week-old mice [(scores in Figure 1B range from ~20-~40 in the 2-week exercise intervention with either no delay (14-0-0) or a 1 week delay (14-7-0,

14-7-2) until testing as compared to D.I. (~0-~10 in the other groups)]. However, in the aging and AD mice (Figure 6D,K) this question remains open, though with vector treatment approximately the same D.I. (~30) is reported as in the adult mice – does the vector reverse a deficit or is the effect similar to exercise in these cohorts?

- “Author response continued: We have found that total exploration time depends on the device used to score the OLM data, although the DI scores do not differ by scoring device. For example, stop watches result in higher overall exploration times, whereas the scoring app used here, we observe lower overall exploration times. Overall exploration times on OLM test are similar to our previous studies where this app was also used to score object exploration (Dong et al., 2022, Keiser et al., 2021, Kwapis et al., 2019, Shu et al., 2018, Kwapis et al., 2018, Vogel-Ciernia et al., 2015). Longer exploration with stop watches is largely due to the requirement to start and stop the watch during each exploration bout. The scoring app that we utilize here does not require a button to be pressed to stop the scoring. Rather, scoring is only counted when the button is actively being pressed, allowing for more precise exploration times that remain consistent with DI scores when using stop watches. Importantly, we avoid scoring mice with too low exploration times by excluding mice from the study if exploration is less than 2 seconds during training or testing. Mice that showed a preference for either object during training ($DI > \pm 20$) were also excluded. ”

- The reviewer now understands that mice were excluded from behavioral testing and analysis if they did not meet certain criteria. Please provide the numbers and group/conditions of mice that were excluded for each experiment. Could these exclusions have been a result of treatment, age or genotype?

4. Re reviewer comment 10. The authors have now updated all 4 habituation graphs to a line graph (Figure S9). They also state that the aged mice traveled a shorter distance.

- To show that the aged mice traveled a shorter distance please equalize y-axes across panels A-D so that the readers can easily see this. Also, please provide the appropriate statistical analysis across ages that indicates there is a difference in distance between young and aged. Were male or female 5xFAD mice used in this study?

5. Re reviewer comment 13. The authors state: “...We have now added additional immunohistochemistry experiments to display the spatial localization of ACVR1C which we observe to be abundant throughout the hippocampus...(see figure S8)”

- Based on this image (Figure S8) ACVR1C is most heavily expressed in the mossy fiber to area CA3 pathway rather than in area CA1. The authors may want to consider additional behavioral tests to interrogate the function of this gene.

6. Re reviewer comment 14, 15. Authors: “For all LTP figures (Figure 1, 4, 5, 6 and S14) we now report the data analyzed across mice/group. We have also included individual data points for each graph. Unfortunately, running distance was not collected for LTP studies. ”

- In the legend of Figure 1D “ LTP is enhanced in male mice following 14 days of exercise (14-0-0) compared 10 to sedentary (0-0-0), ($t(50)=10.74$, $P < 0.0001$), (14-0-0: $n=6$ mice, $n=12$ slices, pooled sedentary 0-0-0: $n=20$ mice, $n=40$ slices).” Please modify to analysis across mice rather than slices.

7. The LTP methods description still lack detail, were recordings made at room temperature or other? How were the recordings analyzed, was there a software program used and are there relevant references? Further, there is no analysis of baseline synaptic transmission across the groups of mice in

this study. Stimulus input - fEPSP slope output graphs should be provided so that running, age, treatment and genotype related changes can be identified. It is also not reported whether the basal fEPSP amplitudes were matched across the groups before LTP induction. These parameters are important, especially given the striking increase in LTP in the 18-month-old FAD mice with ACVR1C-WT vector treatment (Figure S14) to ~160%, comparable adult WT mice.

Reviewer #3 (Remarks to the Author):

The revised ms satisfies all my concerns. The paper is now ready for prime time. It adds significantly to the field and will definitely make an important and lasting impact.

-Tom Carew

Response to Reviews:

We would like to thank the reviewers for their time and continued effort in reviewing this manuscript. We thank Reviewers 1 and 3 for their enthusiastic approval of the first resubmission. In this second resubmission, we address Reviewer 2's comments point-by-point below. Text adjustments based on Reviewer comments are noted in blue within the manuscript.

The authors have improved the manuscript, however, there are remaining concerns that should be addressed:

1. Re reviewer comment 4. The authors have adjusted the discrimination index (D.I.) measurement removing the percentage scores as suggested by the reviewer. However, object location memory (OLM) data does not appear to be consistent across experiments. In Fig. 4D in the AAV control group (EV), in the absence of exercise the D.I. = ~30, suggesting improved OLM. And then in Fig. 5D the exact same treatment, absence of exercise in AAV Control (EV) results in D.I.= ~8 suggesting no effect on OLM. Please explain how the same treatment and testing can result in such vastly different outcomes.

The differences in DI between the EV groups in Figure 4D and 5D are what we would expect given that the training protocol is different in Figure 4D and 5D. In Figure 4, mice are given sufficient time to learn (10 mins, see schematic 4A) and therefore, have high DI scores. In Figure 5D however, mice are given insufficient, *subthreshold*, time to learn (3 mins, see schematic 5A) and therefore, have low DI scores. The range of these DI scores for each training paradigm are in line with previously published work (e.g., Butler et al., 2019; Haettig et al., 2011; Vogel-Ciernia et al., 2015; Keiser et al., 2021; Kwapis et al., 2018; Lopez et al., 2016; Dong et al., 2022; Intlekofer et al., 2013; Kwapis et al., 2019; Shu et al., 2018; McQuown et al., 2011).

2. Re reviewer comment 5. To determine whether *Acvr1c* regulates memory and synaptic plasticity AAV vectors were used to knockdown expression and compared to AAV-EV control (Fig. 4). Further validation of this intervention is provided (Fig. S12) using immunocytochemistry and *Acvr1c* kinase-dead mutant construct and via RT-qPCR. However, it would have been better if each animal that underwent behavioral or electrophysiological testing following vector treatment had undergone histological validation for injection location, spread of vector expression across the extent of the dorsal hippocampus, as such parameters could affect the outcome of behavioral testing.

A minor correction to the comment above, the AAV vectors were not used to knockdown expression. They were used to express a mutant version of *Acvr1c*. With regard to the suggestion to examine post-behavior or physiology, although we would have liked to check the viral expression of each animal as the reviewer suggests, we were unable to check every single animal that went through the behavior as some were randomly assigned for LTP studies that followed. In order to validate with mRNA as well as protein, some brains were randomly assigned for RT-qPCR and others, for immunohistochemistry. That said, all animals checked for viral expression had similar spread in the CA1 region targeted.

3. Re reviewer comment 6. Figure 4E: The panel shows an average of 4s of object exploration. It should be considered that this is a short amount time that may preclude robust interpretations about object discrimination, especially since the test is 5 minutes in duration. Furthermore, it is not clear why the authors do not show the total exploration time for other experiments (e.g. Figure 1).

-“Author response: Thank you for this comment. We have now made clear in the legend

that OLM data in Figure 1 is adapted from Butler et al. (2019) and thus the exploration time is in that paper. “

- Figure 1B in the manuscript represents the findings from Butler et al., 2019, essentially republishing these data to form the premise for a new paper. The reviewer is now referred to that paper to search for the total exploration times and running distances for each behavioral condition.

- The republishing of previous data that was collected with different experimenters at a different point in time and using it as the basis for a new study with the same parameters is very questionable. Indeed, while the same strain and gender of mice was used, the mice in the Butler et al. 2019 study were a different age, 7 weeks old at the time of the start of the exercise intervention and behavioral testing (in Dong et al., 2022, female mice, 8 weeks were utilized), whereas in the current study the adult mice are being tested at 12 weeks of age. Therefore, it is not clear whether the same outcomes could be expected. This experiment should be redone accordingly.

With regard to the comment above about ‘different experimenters’, the first author in the study we reference in Figure 1B (Butler et al., 2019) also performed the exercise and behavior for the RNA-Seq data in this paper (Figure 2). With regard to the comment above about ‘different point in time (age)’, although the age of the mice between our 3 most recent exercise studies (Butler et al., 2019, Dong et al., 2022 and this study) report mouse age differently (Butler et al., 2019 and Dong et al., 2022 report age prior to exercise onset whereas, here, we report age at behavioral test), mice from all studies were ~12 weeks during the OLM training. Thus, the experimenter and age groups are as similar as possible across the two studies.

Although we discuss and display exercise parameters from the Butler et al., 2019 study, utilization of these exercise parameters and examination of cognitive benefit is based on over a decade’s worth of research spearheaded from the Cotman lab (for reference see: Neeper et al., 1996; Neeper et al., 1995; Berchtold et al., 2010; Berchtold et al., 2005; Intlekofer et al., 2013; Adlard et al., 2004; Adlard et al., 2005; Cotman & Berchtold, 2002; Berchtold et al., 2001; Cotman et al., 2007). Given these studies and that we have replicated findings from Butler et al., 2019 in our Dong et al., 2022 (in females) and several other times (including a paper in prep), we did not choose to run this again as the overall effects on behavior are very reproducible and the majority of this manuscript does not focus on exercise, but rather the role of ACVR1C in memory and synaptic plasticity.

- Furthermore, this study includes aging and Alzheimer’s Disease mice. Is OLM affected by aging (e.g. Muria et al., 2007 PMID: 17049363)? Do the same exercise parameters (14-0-0, 14-7-2) modify OLM in the 18-month-old WT and 12-month-old FAD mice? For the adult mice the authors suggest that the effect of ACVR1C-WT vector treatment in 12-week-old mice on D.I. (~25, Figure 5D) is about equal to the effect of exercise in the Butler et al., 2019, in 7-week-old mice [(scores in Figure 1B range from ~20~40 in the 2-week exercise intervention with either no delay (14-0-0) or a 1 week delay (14-7-0, 14-7-2) until testing as compared to D.I. ~0~10 in the other groups)]. However, in the aging and AD mice (Figure 6D,K) this question remains open, though with vector treatment approximately the same D.I. (~30) is reported as in the adult mice – does the vector reverse a deficit or is the effect similar to exercise in these cohorts?

As the reviewer correctly points out OLM is indeed affected by age and in AD-mouse models. We have published a number of papers indicating this and have noted these findings in this

manuscript (For aging mice see pg. 10, lines 13-17, Kwapis 2018a, Kwapis 2018b, Kwapis 2019; for 5xFAD mice see pg. 10, lines 28-29, Forner et al., 2021, also see pg. 10, lines 7-9. Although it is true that exercise parameters are adjusted for studying aging mice vs young mice (e.g. we have observed cognitive benefits from 3 weeks of exercise in aging mice and 2 weeks in young) this question is beyond the scope of this study as we utilized exercise to uncover ACVR1C which we further study in sedentary adult, aging and 5xFAD mice. As the reviewer points out, we are unable to directly compare the effects of ACVR1C wildtype overexpression in aging animals to an exercise experiment conducted in young mice. We also recognize that exercise and ACVR1C manipulations are vastly different modalities likely engaging different mechanisms, circuitry, and systems. In our statement in which we refer to effects of ACVR1C wildtype overexpression in adults (pg. 14, lines 9-12), our intention is to indicate that both exercise and ACVR1C overexpression can facilitate learning in OLM. We have now adjusted our statement to better align with our intention from: “Enhancing ACVR1C (expression or activity), which we find to mimic the cognitive benefits observed with exercise in adult mice” to “which we find to also enable learning as with exercise in adult mice”. Further, as we are not directly comparing aging to young animals in our study and utilize an OLM training period that is subthreshold for learning in aging mice, we discuss the results for aging and 5xFAD mice as enabling learning rather than reversing a deficit. This was addressed in the response to reviewers following the first resubmission as well.

- “Author response continued: We have found that total exploration time depends on the device used to score the OLM data, although the DI scores do not differ by scoring device. For example, stop watches result in higher overall exploration times, whereas the scoring app used here, we observe lower overall exploration times. Overall exploration times on OLM test are similar to our previous studies where this app was also used to score object exploration (Dong et al., 2022, Keiser et al., 2021, Kwapis et al., 2019, Shu et al., 2018, Kwapis et al., 2018, Vogel-Ciernia et al., 2015). Longer exploration with stop watches is largely due to the requirement to start and stop the watch during each exploration bout. The scoring app that we utilize here does not require a button to be pressed to stop the scoring. Rather, scoring is only counted when the button is actively being pressed, allowing for more precise exploration times that remain consistent with DI scores when using stop watches. Importantly, we avoid scoring mice with too low exploration times by excluding mice from the study if exploration is less than 2 seconds during training or testing. Mice that showed a preference for either object during training ($DI > \pm 20$) were also excluded.”

- The reviewer now understands that mice were excluded from behavioral testing and analysis if they did not meet certain criteria. Please provide the numbers and group/conditions of mice that were excluded for each experiment. Could these exclusions have been a result of treatment, age or genotype?

We have now included the number of mice excluded due to low exploration time and include the treatment, age, and genotype (see pg. 21 lines 12-14) “Mice that showed a preference for either object during training ($DI > \pm 20$) were also excluded (n=1 adult ACVR1C-WT, n=1 18 mo. EV and n=1 12 mo. 5xFAD EV were excluded for preference on training and n=1 12 mo. 5xFAD EV was excluded for low exploration on test day)”.

4. Re reviewer comment 10. The authors have now updated all 4 habituation graphs to a line graph (Figure S9). They also state that the aged mice traveled a shorter distance.

- To show that the aged mice traveled a shorter distance please equalize y-axes across panels A-D so that the readers can easily see this. Also, please provide the appropriate

statistical analysis across ages that indicates there is a difference in distance between young and aged. Were male or female 5xFAD mice used in this study?

We have now equalized all y-axes across panels A-D in Figure S9. As we have not directly compared aging vs young mice in the main manuscript text, we have also separated the ages in supplemental Figure S9 to maintain consistency with the main manuscript analyses. Both female and male mice were utilized for the molecular 5xFAD data (Figure 6H), however, only male mice were used for the behavior and LTP (Figure 6I-N), sex is stated in the methods and results sections and indicated within each figure legend. Here, we have provided a graph with analyses indicating lower distance travelled in EV control 18 mo. aging and EV control 12 mo. 5xFAD mice relative to young EV 3 mo. controls.

Total distance traveled during each day of OLM habituation. Distance travelled differed between the ages (Repeated measures Two-way ANOVA: Day: (5,115)=5.16, $P < 0.0001$), Group: (2,23)=3.278, $P = 0.0036$), Interaction: (10,115)=1.76, $P = 0.0760$); Tukey's post hoc test: ** $P < 0.01$, + $P < 0.01$ compared to 3 mo. EV.

5. Re reviewer comment 13. The authors state: "...We have now added additional immunohistochemistry experiments to display the spatial localization of ACVR1C which we observe to be abundant throughout the hippocampus (see figure S8)"

- Based on this image (Figure S8) ACVR1C is most heavily expressed in the mossy fiber to area CA3 pathway rather than in area CA1. The authors may want to consider additional behavioral tests to interrogate the function of this gene.

In this study, we chose to target area CA1 to enhance or disrupt ACVR1C function as the behavior and LTP is dependent on this region. We agree with the reviewer that it appears to be that ACVR1C is more heavily expressed in the mossy fiber to CA3 pathway and agree that it would be interesting for future studies to study the role of ACVR1C in CA3-dependent mechanisms and memory processes.

6. Re reviewer comment 14, 15. Authors: "For all LTP figures (Figure 1, 4, 5, 6 and S14) we now report the data analyzed across mice/group. We have also included individual data points for each graph. Unfortunately, running distance was not collected for LTP studies."

- In the legend of Figure 1D “ LTP is enhanced in male mice following 14 days of exercise (14-0-0) compared 10 to sedentary (0-0-0), ($t(50)=10.74$, $P < 0.0001$), (14-0-0: $n=6$ mice, $n=12$ slices, pooled sedentary 0-0-0: $n=20$ mice, $n=40$ slices).” Please modify to analysis across mice rather than slices.

We have now modified the analysis across mice instead of slices. We thank the reviewer for pointing out this.

7. The LTP methods description still lack detail, were recordings made at room temperature or other? How were the recordings analyzed, was there a software program used and are there relevant references? Further, there is no analysis of baseline synaptic transmission across the groups of mice in this study. Stimulus input - fEPSP slope output graphs should be provided so that running, age, treatment and genotype related changes can be identified. It is also not reported whether the basal fEPSP amplitudes were matched across the groups before LTP induction. These parameters are important, especially given the striking increase in LTP in the 18-month-old FAD mice with ACVR1C-WT vector treatment (Figure S14) to ~160%, comparable adult WT mice.

Information regarding recording temperature for slides can be found on pg. 21 line 26 “ 31 ± 10 C”. Additional methods details and a reference on the software program requested by the reviewer have now been added to the methods section (see pg. 22, lines 5-6) “Data were collected and digitized by NAC 2.0 Neurodata Acquisition System (Theta Burst)”. Unfortunately, analyses of baseline synaptic transmission were not collected for this study. The basal fEPSP slope was normalized across the groups before LTP induction and this information is included in the methods section (see pg. 22 lines 7-9) “The fEPSP slope was measured at 10–90% fall of the slope and data in figures on LTP were normalized to the last 20 min of baseline”.

REVIEWER COMMENTS

Reviewer #2 (Remarks to the Author):

The authors have improved the manuscript. However, the following questions and comments should be addressed:

1. Previous reviewer comment 7. The LTP methods description still lack detail,.....especially given the striking increase in LTP in the 18-month-old FAD mice with ACVR1C-WT vector treatment (Figure S14) to ~160%, comparable adult WT mice.

Reviewer: the authors did not address the issue pertaining to the magnitude of the potentiation in the aged mice. Indeed, the LTP magnitude is quite similar (~140% - ~160%) across experiments in adult, aged and Alzheimer's disease (5xFAD) mice.

Figure 6. In panel 6F the scale bar (mV/ms) is missing. Please add this information. In panel 6M the scale (1 mV/5 ms) is shown and indicates that the amplitude of the response is ~3 mV. This seems to be rather large for 18-month-old C57 mice.

Indeed, the area CA1 LTP responses in old mice are generally reported to be smaller than in young animals (e.g. Watson et al., 2006 <https://doi.org/10.1002/jnr.21040>). However, in this manuscript the amplitude in slices derived the young adult mice (~2mV) in Figure 1D-1E-1F and Figure 4G is lower than in aged mice.

Figure S14. The scale bar (mV/ms) for the LTP data in 18-month-old Alzheimer's Disease model mice (5xFAD) in panel S14A is missing. Please add the scale bar to this figure.

2. The representation of the novel object data: Page 21. Lines 6-8 "Total exploration time was recorded (t) and preference for the novel object was expressed as discrimination index ($DI = (t_{\text{novel}} - t_{\text{familiar}}) / (t_{\text{novel}} + t_{\text{familiar}}) \times 100$)."

- The above calculation has been used in the papers published by the authors' laboratory. As described in the papers cited below using exactly the same formula, this measure is a relative discrimination value or discrimination index, which is not influenced by differences in exploration time. All values are between -1 and +1.

Please remove the multiplication by 100 of the data and modify all the figures accordingly.

- Alternatively, calculate the recognition or preference index. This is the time spent exploring the novel object divided by the total time. All values will fall between 0 and 1. It is often multiplied by 100 and used as percentage value (Lueptow, 2017).

Antunes and Biala, 2012, PMID: 22160349 doi: 10.1007/s10339-011-0430-z and Lueptow, 2017
URL:<https://www.jove.com/video/55718>
DOI: doi:10.3791/55718)

3. Reviewer comment on Figure 1B: Figure 1B in the manuscript represents the findings from Butler et al., 2019, essentially republishing these data to form the premise for a new paper. The reviewer is now referred to that paper to search for the total exploration times and running distances for each behavioral condition...The republishing of previous data....'

Author response: "Although we discuss and display exercise parameters from the Butler et al., 2019

study, utilization of these exercise parameters and examination of cognitive benefit is based on over a decade's worth of research spearheaded from the Cotman lab (for reference see: Neeper et al., 1996; Neeper et al., 1995; Berchtold et al., 2010; Berchtold et al., 2005; Intlekofer et al., 2013; Adlard et al., 2004; Adlard et al., 2005; Cotman & Berchtold, 2002; Berchtold et al., 2001; Cotman et al., 2007). Given these studies and that we have replicated findings from Butler et al., 2019 in our Dong et al., 2022 (in females) and several other times (including a paper in prep), we did not choose to run this again as the overall effects on behavior are very reproducible and the majority of this manuscript does not focus on exercise, but rather the role of ACVR1C in memory and synaptic plasticity."

Reviewer: As the data in Figure 1B is reproduced and reorganized from a previous paper it cannot be considered original research. This information is more appropriate for a review article. It also gives the impression that the LTP data reported in the same figure (Figure 1D-G) was derived from the same animals for which behavior is reported. Suggest performing the experiment de novo or removing the current panel from the paper.

4. Initial review 6. Re reviewer comment 14, 15. Authors: "For all LTP figures (Figure 1, 4, 5, 6 and S14) we now report the data analyzed across mice/group. We have also included individual data points for each graph. Unfortunately, running distance was not collected for LTP studies. "

In the legend of Figure 1D " LTP is enhanced in male mice following 14 days of exercise (14-0-0) compared 10 to sedentary (0-0-0), (t(50)=10.74, P <0.0001), (14-0-0: n=6 mice, n=12 slices, pooled sedentary 0-0-0: 11 n=20 mice, n=40 slices)." Please modify to analysis across mice rather than slices.

Response: We have now modified the analysis across mice instead of slices. We thank the reviewer for pointing this out."

Reviewer: The statistical analysis of the LTP data should be checked in detail.

Legend of Figure 1:

"(D) LTP is enhanced in male mice following 14 days of exercise (14-0-0) compared to sedentary (0-0-0), (t(24)=9.78, P <0.0001), (14-0-0: n=6 mice, n=12 slices, pooled sedentary 0-0-0: n=20 mice, n=40 slices). "

Reviewer: The design of the experiment is unbalanced, why were 20 sedentary control mice used?

"(E) LTP remains elevated following a 7-day delay (14-7-0) and 2 days of reactivating exercise (14-7-2) does not further potentiate responses, One-way ANOVA, Group: (F(2,53) = 34.40, P < 0.0001). Tukey's post hoc test, 50-60 min post TBS: ****P <0.0001 for both 14-7-0 and 14-7-2, relative to 0-0-0. (14-7-0: n=4 mice, n=8 slices, 14-7-2: n=4 mice, n=8 slices). "

Reviewer: If there are 4 mice (14-7-0) + 4 mice (14-7-2) and + 20 mice from sedentary group = 28 mice, why does the ANOVA reflect >50 animals (F(2,53) = 34.40, P < 0.0001).?

"(F) LTP returns to baseline following a 14-day break (14-14-0) and 2 days of re-introduction to exercise (14-14-2) is sufficient to re-gain elevated LTP, One-way ANOVA, Group: (F(2,53) = 34.40, P < 0.0001). Tukey's post hoc test, 50-60 min post TBS: ****P <0.0001 for 14-14-2 relative to 0-0-0, *P <0.05 for 14-14-2 relative to 14-14-0. (14-14-0: n=5 mice, n=10 slices, 14-14-2: n=5 mice, n=10 slices, 0-0-2: n=5 mice, n=10 slices)."

Reviewer: Similar concern pertaining to the analysis. Mice (n): 5+5+5+20 =35, however, (F(2,53) = 34.40, P < 0.0001).

Also the F scores are identical for panels E and F, please check.

Legend of Figure 5:

"Figure 5G, LTP time course (n=5 mice/group, n=10 slices/group). TBS applied at arrow. (H) Mean level of potentiation 50-60 min post TBS showing enhanced LTP maintenance in WT-infused mice. ACVR1C overexpression enhances LTP relative to EV control (t(6)=3.510, P =0.0127). "

Reviewer: Mice n=5 per group is reported, however in Figure 5H, statistical value t(6).. and the histogram shows EV (n=3) and AVRC1-WT (n=5).

Were 2 animals were excluded from this analysis? Please provide the inclusion and exclusion criteria.

Legend of Figure 6:

"(F) LTP time course (n=3 mice/group, n=6 slices/condition). TBS applied at arrow. (G) ACVR1C overexpression enhances LTP in 18 mo. C57BL6/J mice (t(6)=3.540, P =0.0122)."

Reviewer: Mice n=3 per group (6 total). The t(6).. score does not match sample size and panel 6G shows EV (n=5) and AVR1C-WT (n=3).

"(M) LTP time course (n=4 mice/group, n=6 slices/condition). TBS applied at arrow. (N) ACVR1C overexpression enhances LTP in 12 mo. 5xFAD mice relative to EV control (t(7)=3.852, P =0.0063)."

Reviewer: Mice n=4 per group. The t(7) score does not match sample size and panel 6N shows EV (n=4) and AVR1C-WT (n=5).

Response to Referees:

We would like to thank the reviewer for their time, comments and continued effort which has helped to strengthen the manuscript. In this third resubmission, we address Reviewer 2's comments point-by-point below. **Text adjustments for this resubmission based on Reviewer comments are noted in purple within the manuscript.** We still included the revisions from the first review in red and the second review in blue for completeness.

Reviewer #2 (Remarks to the Author):

The authors have improved the manuscript. However, the following questions and comments should be addressed:

1. Previous reviewer comment 7. The LTP methods description still lack detail,.....especially given the striking increase in LTP in the 18-month-old FAD mice with ACVR1C-WT vector treatment (Figure S14) to ~160%, comparable adult WT mice.

Reviewer: the authors did not address the issue pertaining to the magnitude of the potentiation in the aged mice. Indeed, the LTP magnitude is quite similar (~140% - ~160%) across experiments in adult, aged and Alzheimer's disease (5xFAD) mice.

Interface recording chambers (used in our study) are well known for producing healthy slices that can produce larger fEPSP when compared to fEPSP collected in a submerged chamber. Equally important is that we don't see any noticeable change in field response size until 22-24 months of age. This is similar to what is observed in the Watson et al., 2006 study cited by the reviewer below, which examines 20-26 month old mice. It is also important to consider strain background, sex, LTP induction protocol, etc. Our aging mice are 18 months old, and we are still within the window of producing normal sized fEPSPs found in adult slices.

For the Reviewer's convenience, here are a couple of our recent publications using 18 mo. mice and you can see that the response sizes are very similar to 18 mo. controls as reported in the present manuscript.

- (1) Javonillo et al., 2022 Front Neurosci. 2022 Jan 24;15:785276. doi: 10.3389/fnins.2021.785276.
- (2) Baglietto-Vargas et al., 2021 Nat Commun. Apr 23;12(1):2421. doi: 10.1038/s41467-021-22624-z.

Figure 6. In panel 6F the scale bar (mV/ms) is missing. Please add this information. In panel 6M the scale (1 mV/5 ms) is shown and indicates that the amplitude of the response is ~3 mV. This seems to be rather large for 18-month-old C57 mice.

We thank the reviewer for catching the missing scale bar. We have now added it.

We would like to remind the reviewer that slope, not amplitude, was measured in all LTP experiments. The slope of the response is the most accurate measure of LTP, and the associated amplitude of that response can be either small or large depending how close you are recording from the cell body layer. The level of potentiation that is achieved in any LTP experiment is always based on the difference in baseline and potentiated slope, not amplitude.

Additionally, as a point of clarification, panel 6M is showing 12 mo. (not 18 mo.) 5xFAD mice injected with the empty vector (top circle) or with the wildtype form of *Acvr1c* (bottom circle). Both groups shown in 6M are 5xFAD mice (there is no C57 group alone). Perhaps the

confusion is that the 5xFAD mice are receiving the AAV overexpressing wildtype Acvr1c (denoted as ACVR1C-WT / 12 mo. 5xFAD).

Indeed, the area CA1 LTP responses in old mice are generally reported to be smaller than in young animals (e.g. Watson et al., 2006 <https://doi.org/10.1002/jnr.21040>). However, in this manuscript the amplitude in slices derived the young adult mice (~2mV) in Figure 1D-1E-1F and Figure 4G is lower than in aged mice.

We apologize for having inconsistent scaling for the represented traces in the last version of the manuscript. We have gone through each set of traces for all LTP experiments and made sure that the scaling was the same throughout the revised manuscript.

In addition, it should be noted that we are not directly comparing aging vs young animals for any parameters in our study. However, as a point of interest, we have assessed whether differences exist in the baseline fEPSP amplitude in slices among young vs aging mice. Please refer to the figure below.

The above figure plots baseline amplitude of each slice that was used in our recordings from all 18 mo. mice in Fig 6 and Supp. 14 and compared them to slice recordings from young adult mice that were used in Fig 1 and 4G. All the amplitudes measured from these experiments were ran in the same year. As one can see from the above plot, and inside the descriptive statistics box, there was no significant difference in amplitude between ages (t-test, $p = 0.903$).

Figure S14. The scale bar (mV/ms) for the LTP data in 18-month-old Alzheimer's Disease model mice (5xFAD) in panel S14A is missing. Please add the scale bar to this figure.

We thank the reviewer for catching the missing scale bar. We have now added it.

2. The representation of the novel object data: Page 21. Lines 6-8 "Total exploration time was recorded (t) and preference for the novel object was expressed as discrimination

index ($DI = (tnovel - tfamiliar) / (tnovel + tfamiliar) \times 100$).”

- The above calculation has been used in the papers published by the authors' laboratory. As described in the papers cited below using exactly the same formula, this measure is a relative discrimination value or discrimination index, which is not influenced by differences in exploration time. All values are between -1 and +1.

Please remove the multiplication by 100 of the data and modify all the figures accordingly.

- Alternatively, calculate the recognition or preference index. This is the time spent exploring the novel object divided by the total time. All values will fall between 0 and 1. It is often multiplied by 100 and used as percentage value (Lueptow, 2017).

Antunes and Biala, 2012, PMID: 22160349 doi: 10.1007/s10339-011-0430-z and Lueptow, 2017 URL: <https://www.jove.com/video/55718>
DOI: doi:10.3791/55718)

We thank the reviewer for providing this reference. Below we show an example comparison between using the approach we and others have used in numerous studies (left panel) compared to the preference index from the Lueptow 2017 J Vis Exp paper (right panel). The left panel uses $DI = (tnovel - tfamiliar) / (tnovel + tfamiliar) \times 100$. The right panel uses $PI = (tnovel) / (tnovel + tfamiliar) \times 100$. The statistical results are identical in either case. This comes down to preference in how data is displayed. We would like to keep our DI display because it allows us to be consistent with all of our other studies and it gives an easy visualization regarding preference for the novel object location (above zero) versus preference for the original object location (below zero).

(A) Discrimination Index. Disrupting ACVR1C function (ACVR1C-KD) leads to impaired OLM ($t(17)=4.65$, $P=0.0002$). (B) Preference Index. Disrupting ACVR1C function (ACVR1C-KD) leads to impaired OLM ($t(17)=4.65$, $P=0.0002$).

3. Reviewer comment on Figure 1B: Figure 1B in the manuscript represents the findings from Butler et al., 2019, essentially republishing these data to form the premise for a new paper. The reviewer is now referred to that paper to search for the total exploration times

and running distances for each behavioral condition...The republishing of previous data.....'

Author response: "Although we discuss and display exercise parameters from the Butler et al., 2019 study, utilization of these exercise parameters and examination of cognitive benefit is based on over a decade's worth of research spearheaded from the Cotman lab (for reference see: Neeper et al., 1996; Neeper et al., 1995; Berchtold et al., 2010; Berchtold et al., 2005; Intlekofer et al., 2013; Adlard et al., 2004; Adlard et al., 2005; Cotman & Berchtold, 2002; Berchtold et al., 2001; Cotman et al., 2007). Given these studies and that we have replicated findings from Butler et al., 2019 in our Dong et al., 2022 (in females) and several other times (including a paper in prep), we did not choose to run this again as the overall effects on behavior are very reproducible and the majority of this manuscript does not focus on exercise, but rather the role of ACVR1C in memory and synaptic plasticity."

Reviewer: As the data in Figure 1B is reproduced and reorganized from a previous paper it cannot be considered original research. This information is more appropriate for a review article. It also gives the impression that the LTP data reported in the same figure (Figure 1D-G) was derived from the same animals for which behavior is reported. Suggest performing the experiment de novo or removing the current panel from the paper.

The reviewer writes that the adapted data shown in Figure 1B is being 'considered original research.' That is not correct. We have clearly explained that Figure 1B is adapted from our Butler et al., 2019 paper and illustrated in a manner that more easily explains the effect of different exercise parameters on *sub-threshold* learning to provide rationale for why we are using these exercise parameters in this study. We are also explicitly clear that the LTP data is derived from new cohorts of animals (LTP data in Figure 1 comes from animals post-training, therefore not possible to be from the Butler study where animals are trained and tested). Taking data and presenting it for rationale does not violate the journal guidelines as long as it is represented correctly (which it is here). Further, the website of the original journal for the Butler et al. paper states that we have permission to reproduce figures see point 3: <https://learnmem.cshlp.org/site/misc/terms.xhtml>. Additionally, to ensure reproduction was allowed, we have recently obtained written approval from the assistant editor of *Learning & Memory*, Dr. Susan Cushman that formal permission is not needed for reproduction. Dr. Cushman instructed us to write that "(Fig 1B; data taken from our previous study Butler et al., 2019)". Thus, with all due respect to the reviewer, we would like to keep Figure 1B as it is crucial for explaining the rationale behind the LTP and the RNAseq experiments.

4. Initial review 6. Re reviewer comment 14, 15. Authors: "For all LTP figures (Figure 1, 4, 5, 6 and S14) we now report the data analyzed across mice/group. We have also included individual data points for each graph. Unfortunately, running distance was not collected for LTP studies. "

In the legend of Figure 1D " LTP is enhanced in male mice following 14 days of exercise (14-0-0) compared 10 to sedentary (0-0-0), ($t(50)=10.74$, $P < 0.0001$), (14-0-0: $n=6$ mice, $n=12$ slices, pooled sedentary 0-0-0: 11 $n=20$ mice, $n=40$ slices)." Please modify to analysis across mice rather than slices.

Response: We have now modified the analysis across mice instead of slices. We thank the reviewer for pointing this out."

Reviewer: The statistical analysis of the LTP data should be checked in detail.

We thank the reviewer for noting this. In the last submission we updated the analysis in the figure legend presented under the figure, but accidentally left out the updated analysis in the figure legends following the results section. Both are now updated in the revised manuscript.

Legend of Figure 1:

“(D) LTP is enhanced in male mice following 14 days of exercise (14-0-0) compared to sedentary (0-0-0), ($t(24)=9.78$, $P < 0.0001$), (14-0-0: $n=6$ mice, $n=12$ slices, pooled sedentary 0-0-0: $n=20$ mice, $n=40$ slices). “

Reviewer: The design of the experiment is unbalanced, why were 20 sedentary control mice used?

During each LTP experiment that involved an exercise mouse we ensured that recordings were also taken from sedentary control mice on the same day and time frame. Because we did not find significant differences in LTP among any of the control slices from each of the four experiments, we pooled the data. Pooling the control data made it easier to present the averaged data shown in the histogram for all groups (to avoid showing a control bar for every single group). However, we see how that could lead to some confusion as graphed (especially with the revised figure), so we separated out the controls for each experiment and re-analyzed and replotted the original controls that were ran with each of the exercise studies.

“(E) LTP remains elevated following a 7-day delay (14-7-0) and 2 days of reactivating exercise (14-7-2) does not further potentiate responses, One-way ANOVA, Group: ($F(2,53) = 34.40$, $P < 0.0001$). Tukey’s post hoc test, 50-60 min post TBS: ** $P < 0.0001$ for both 14-7-0 and 14-7-2, relative to 0-0-0. (14-7-0: $n=4$ mice, $n=8$ slices, 14-7-2: $n=4$ mice, $n=8$ slices). “**

Reviewer: If there are 4 mice (14-7-0) + 4 mice (14-7-2) and + 20 mice from sedentary group = 28 mice, why does the ANOVA reflect >50 animals ($F(2,53) = 34.40$, $P < 0.0001$).?

“(F) LTP returns to baseline following a 14-day break (14-14-0) and 2 days of re-introduction to exercise (14-14-2) is sufficient to re-gain elevated LTP, One-way ANOVA, Group: ($F(2,53) = 34.40$, $P < 0.0001$). Tukey’s post hoc test, 50-60 min post TBS: ** $P < 0.0001$ for 14-14-2 relative to 0-0-0, * $P < 0.05$ for 14-14-2 relative to 14-14-0. (14-14-0: $n=5$ mice, $n=10$ slices, 14-14-2: $n=5$ mice, $n=10$ slices, 0-0-2: $n=5$ mice, $n=10$ slices).”**

Reviewer: Similar concern pertaining to the analysis. Mice (n): $5+5+5+20 = 35$, however, ($F(2,53) = 34.40$, $P < 0.0001$).

We thank the reviewer for noting this. It has been corrected in the revised manuscript.

Also the F scores are identical for panels E and F, please check.

We thank the reviewer for noting this and have now updated the F scores.

Legend of Figure 5:

“Figure 5G, LTP time course (n=5 mice/group, n=10 slices/group). TBS applied at arrow. (H) Mean level of potentiation 50-60 min post TBS showing enhanced LTP maintenance in WT-infused mice. ACVR1C overexpression enhances LTP relative to EV control (t(6)=3.510, P =0.0127). “

Reviewer: Mice n=5 per group is reported, however in Figure 5H, statistical value t(6).. and the histogram shows EV (n=3) and AVRC1-WT (n=5).

Were 2 animals were excluded from this analysis? Please provide the inclusion and exclusion criteria.

We thank the reviewer for noting this and have now updated the number of mice and slices in the legend to include both groups.

Legend of Figure 6:

“(F) LTP time course (n=3 mice/group, n=6 slices/condition). TBS applied at arrow. (G) ACVR1C overexpression enhances LTP in 18 mo. C57BL6/J mice (t(6)=3.540, P =0.0122).”

Reviewer: Mice n=3 per group (6 total). The t(6).. score does not match sample size and panel 6G shows EV (n=5) and AVR1C-WT (n=3).

We thank the reviewer for noting this and have now updated the number of mice and slices in the legend to include both groups.

“(M) LTP time course (n=4 mice/group, n=6 slices/condition). TBS applied at arrow. (N) ACVR1C overexpression enhances LTP in 12 mo. 5xFAD mice relative to EV control (t(7)=3.852, P =0.0063).”

Reviewer: Mice n=4 per group. The t(7) score does not match sample size and panel 6N shows EV (n=4) and AVR1C-WT (n=5).

We thank the reviewer for noting this and have now updated the number of mice and slices in the legend to include both groups.

REVIEWERS' COMMENTS

Reviewer #2 (Remarks to the Author):

The authors have further improved the paper.